# Neural Lighting Simulation for Urban Scenes

**Ava Pun**[1,3][*][†]   **Gary Sun**[1,3][*][†]   **Jingkang Wang**[1,2][*]   **Yun Chen**[1,2]   **Ze Yang**[1,2]
**Sivabalan Manivasagam**[1,2]   **Wei-Chiu Ma**[1,4]   **Raquel Urtasun**[1,2]

Waabi[1]   University of Toronto[2]   University of Waterloo[3]   MIT[4]

## Abstract

Different outdoor illumination conditions drastically alter the appearance of urban scenes, and they can harm the performance of image-based robot perception systems if not seen during training. Camera simulation provides a cost-effective solution to create a large dataset of images captured under different lighting conditions. Towards this goal, we propose *LightSim*, a neural lighting camera simulation system that enables diverse, realistic, and controllable data generation. LightSim automatically builds *lighting-aware digital twins* at scale from collected raw sensor data and decomposes the scene into dynamic actors and static background with accurate geometry, appearance, and estimated scene lighting. These digital twins enable actor insertion, modification, removal, and rendering from a new viewpoint, all in a lighting-aware manner. LightSim then combines physically-based and learnable deferred rendering to perform realistic relighting of modified scenes, such as altering the sun location and modifying the shadows or changing the sun brightness, producing spatially- and temporally-consistent camera videos. Our experiments show that LightSim generates more realistic relighting results than prior work. Importantly, training perception models on data generated by Light-Sim can significantly improve their performance. Our project page is available at https://waabi.ai/lightsim/.

## 1   Introduction

Humans can perceive their surroundings under different lighting conditions, such as identifying traffic participants while driving on a dimly lit road or under mild sun glare. Unfortunately, modern camera-based perception systems, such as those in self-driving vehicles (SDVs), are not as robust [52, 32]. They typically only perform well in the canonical setting they were trained in, and their performance drops significantly under different unseen scenarios, such as in low-light conditions [52, 59, 32]. To reduce distribution shift, we could collect data under various lighting conditions for each area in which we want to deploy the SDVs, generating a diverse dataset on which to train the perception system. Unfortunately, this is not scalable as it is too expensive and time-consuming.

Simulation is a cost-effective alternative for generating large-scale data with diverse lighting. To be effective, a simulator should be realistic, controllable, and diverse. Realistic simulation of camera data enables generalization to the real world. Controllable actor placement and lighting allow the simulator to generate the desired training scenarios. Diverse backgrounds, actor assets, and lighting conditions allow simulation to cover the full real-world distribution. While existing game-engine-based self-driving simulators such as CARLA [18, 68] are controllable, they provide a limited number of manually designed assets and lighting conditions. Perception systems trained on this data generalize poorly to the real world [26, 1, 61].

---

[*]Equal contributions.
[†]Work done while a research intern at Waabi.

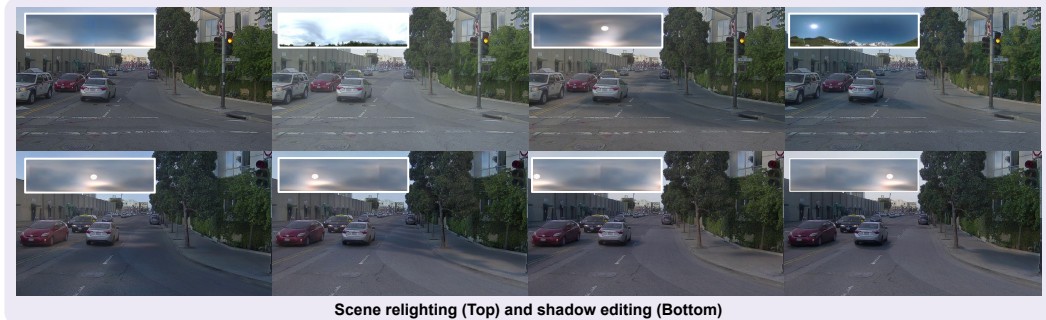

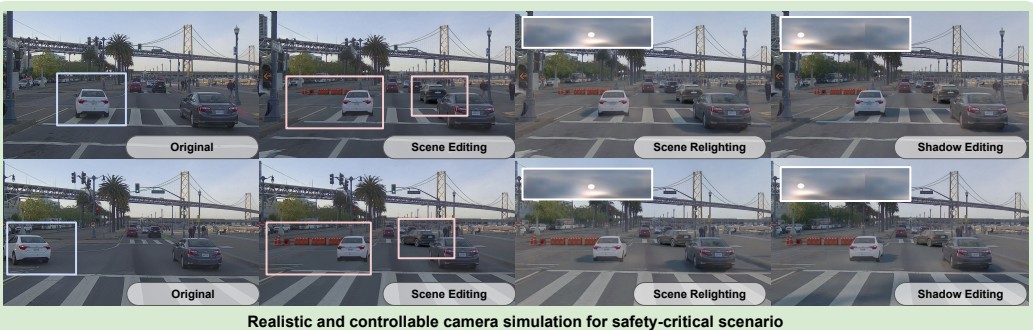

Figure 1: **LightSim builds digital twins from large-scale data with lighting variations and generates high-fidelity simulation videos.** Top: LightSim produces realistic scene relighting and shadow editing videos. Bottom: We generate a safety-critical scenario with two vehicles cutting in and perform lighting-aware camera simulation. See Appendix E and project page for more examples.

To build a more realistic, scalable, and diverse simulator, we instead propose reconstructing "digital twins" of the real world at scale from a moving platform equipped with LiDAR and cameras. By reconstructing the geometry and appearance of the actors and static backgrounds to create composable neural assets, as well as estimating the scene lighting, the digital twins can provide a rich asset library for simulation. Existing methods for building digital twins and data-driven simulators [2, 40, 89, 73, 57] bake the lighting into the scene, making simulation under new lighting conditions impossible. In contrast, we create lighting-aware digital twins, which enable actor insertion, removal, modification, and rendering from new viewpoints with accurate illumination, shadows, occlusion, and temporal consistency. Moreover, by estimating the scene lighting, we can use the digital twins for relighting.

Given a digital twin, we must relight the scene to a target novel lighting condition. This is, however, a challenging task. Prior image-based synthesis methods [4, 5] perform relighting via 2D style transfer techniques, but they typically lack human-interpretable controllability, are not temporally or spatially consistent, and can have artifacts. Inverse neural rendering methods [82, 49, 39, 83] aim to decompose the scene into geometry, materials and lighting, which allows for relighting through physically-based rendering [7, 30]. However, the ill-posed nature of lighting estimation and intrinsic decomposition makes it challenging to fully disentangle each aspect accurately, resulting in unrealistic relit images. Both approaches, while successful on synthetic scenes or outdoor landmarks [51] with dense data captured under many lighting conditions, have difficulty performing well on large outdoor scenes. This is primarily due to the scarcity of real-world scenes captured under different lighting conditions. Moreover, most prior works perform relighting on static scenes and have not demonstrated realistic relighting for dynamic scenes where both the actors and the camera viewpoint are changing, resulting in inter-object lighting effects that are challenging to simulate.

In this paper, we present *LightSim*, a novel lighting simulation system for urban driving scenes that generates diverse, controllable, and realistic camera data. To achieve *diversity*, LightSim reconstructs lighting-aware digital twins from real-world sensor data, creating a large library of assets and lighting environment maps for simulation. LightSim then leverages physically-based rendering to enable *controllable* simulation of the dynamic scene, allowing for arbitrary actor placement, SDV location, and novel lighting conditions. To improve realism, we further enhance the physically-based renderer

with an image-based neural deferred renderer to perform relighting, enabling data-driven learning to overcome geometry artifacts and ambiguity in the decomposition due to not having perfect knowledge of the scene and sensor configuration. To overcome the lack of real-world scenes captured under different lighting conditions, we train our neural deferred renderer on a mixture of real and synthetic data generated with physically-based rendering on our reconstructed digital twins.

We demonstrate the effectiveness of our approach on PandaSet [87], which contains more than 100 real-world self-driving scenes covering 20 unique kilometers. LightSim significantly outperforms the state of the art, producing high-quality photorealistic driving videos under a wide range of lighting conditions. We then showcase several capabilties of LightSim, such as its ability to create high-fidelity and lighting-aware actor insertion, scene relighting, and shadow editing (Fig. 1). We demonstrate that by training camera perception models with LightSim-generated data, we can achieve significant performance improvements, making the models more robust to lighting variation. We believe LightSim is an important step towards enhancing the safety of self-driving development and deployment.

## 2   Related Work

**Outdoor lighting estimation:**   As a first step for lighting simulation, lighting estimation aims to recover the 360° HDR light field from observed images for photo-realistic virtual object insertion [20, 28, 19, 27, 37, 96, 67, 71, 101, 82, 78, 74, 83]. Existing works generally use neural networks to predict various lighting representations, such as environment maps [71, 101, 78, 82, 74, 83], spherical lobes [8, 43], light probes [37], and sky models [28, 27, 96], from a single image. For outdoor scenes, handling the high dynamic range caused by the presence of the sun is a significant challenge. Moreover, due to the scarcity of real-world datasets and the ill-posed nature of lighting estimation [82, 74], it is challenging to precisely predict peak intensity and sun location from a single limited field-of-view low-dynamic range (LDR) image. To mitigate these issues, SOLDNet [101, 74] enhances the diversity of material and lighting conditions with synthetic data and introduces a disentangled global and local lighting latent representation to handle spatially-varying effects. NLFE [82] uses hybrid sky dome / light volume and introduces adversarial training to improve realism with differentiable actor insertion. In contrast, our work fully leverages the sensory data available in a self-driving setting (*i.e.*, multi-camera images, LiDAR, and GPS) to accurately recover spatially-varying environment maps.

**Inverse rendering with lighting:**   Another way to obtain lighting representations is through joint geometry, material, and lighting optimization with inverse rendering [67, 42, 84, 55, 79, 23, 49, 91, 64, 39, 41, 92, 83]. However, since optimization is conducted on a single scene and material decomposition is the primary goal, the optimized lighting representations are usually not generalizable and do not work well for relighting [67, 55, 23]. Inspired by the success of NeRF in high-fidelity 3D reconstruction, some works use volume rendering to recover material and lighting given image collections under different lighting conditions [64, 49, 39], but they are limited in realism and cannot generalize to unseen scenes. Most recently, FEGR [83], independent and concurrent to our work, proposes a hybrid framework to recover scene geometry, material and HDR lighting of urban scenes from posed camera images. It demonstrates realistic lighting simulation (actor insertion, shadow editing, and scene relighting) for static scenes. However, due to imperfect geometry and material/lighting decomposition, relighting in driving scenes introduces several noticeable artifacts, including unrealistic scene color, blurry rendering results that miss high-frequency details, obvious mesh boundaries on trees and buildings, and unnatural sky regions when using other HDR maps. In contrast, LightSim learns on many driving scenes and performs photorealistic enhanced deferred shading to produce more realistic relighting videos for *dynamic scenes*.

**Camera simulation for robotics:**   There is extensive work on developing simulated environments for safer and faster robot development [33, 86, 53, 17, 75, 38, 15, 29, 17, 35, 9, 13]. Two major lines of work in sensor simulation for self-driving include graphics-based [18, 68] and data-driven simulation [80, 66, 50, 88]. Graphics-based simulators like CARLA [18] and AirSim [68] are fast and controllable, but they face limitations in scaling and diversity due to costly manual efforts in asset building and scenario creation. Moreover, these approaches can generate unrealistic sensor data that have a large domain gap for autonomy [26, 85]. Data-driven approaches leverage computer vision techniques and real-world data to build simulators for self driving [34, 3, 2, 81, 40]. Unfortunately, existing works tend to fall short of realism, struggle with visual artifacts and domain gap [34, 3, 2, 81,

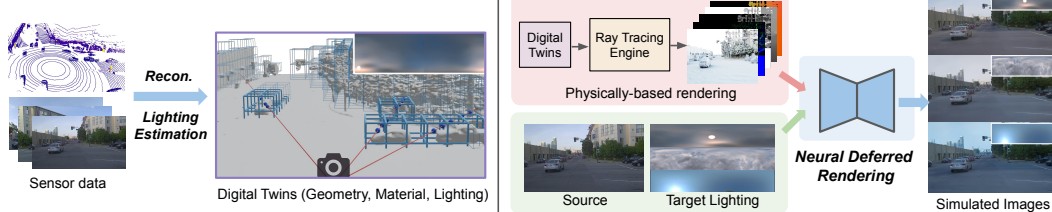

Figure 2: **Overview of LightSim.** Given sensor observations of the scene, we first perform neural scene reconstruction and lighting estimation to build lighting-aware digital twins (**left**). Given a target lighting, we then perform both physically-based and neural deferred rendering to simulate realistic driving videos under diverse lighting conditions (**right**).

40], and lack comprehensive control in synthesizing novel views [12, 82, 79, 90, 72]. Most recent works [57, 36, 77, 89, 46] use neural radiance fields to build digital twins and represent background scenes and agents as MLPs, enabling photorealistic rendering and controllable simulation across a single snippet. However, these works bake lighting and shadows into the radiance field and therefore cannot conduct actor insertion under various lighting conditions or scene-level lighting simulation. In contrast, LightSim builds lighting-aware digital twins for more controllable camera simulation.

# 3 Building Lighting-Aware Digital Twins of the Real World

The goal of this paper is to create a diverse, controllable, and realistic simulator that can generate camera data of scenes at scale under diverse lighting conditions. Towards this goal, LightSim first reconstructs lighting-aware digital twins from camera and LiDAR data collected by a moving platform. The digital twin comprises the geometry and appearance of the static background and dynamic actors obtained through neural rendering (Sec. 3.1), as well as the estimated scene lighting (Sec. 3.2). We carefully build this representation to allow full controllability of the scene, including modifying actor placement or SDV position, adding and removing actors in a lighting-aware manner for accurate shadows and occlusion, and modifying lighting conditions, such as changing the sun's location or intensity. In Sec. 3.3, we then describe how we perform learning on sensor data to build the digital twin and estimate lighting. In Sec. 4, we describe how we perform realistic scene relighting with the digital twin to generate the final temporally consistent video.

## 3.1 Neural Scene Reconstruction

Inspired by [89, 77, 57], we learn the scene geometry and base texture via neural fields. We design our neural field $F : \mathbf{x} \mapsto (s, \mathbf{k}_d)$ to map a 3D location $\mathbf{x}$ to a signed distance $s \in \mathbb{R}$ and view-independent diffuse color $\mathbf{k}_d \in \mathbb{R}^3$. We decompose the driving scene into a static background $\mathcal{B}$ and a set of dynamic actors $\{\mathcal{A}_i\}_{i=1}^M$ and map multi-resolution spatial feature grids [54] using two MLP networks: one for the static scene and one for the dynamic actors. This compositional representation allows for 3D-aware actor insertion, removal, or manipulation within the background. From our learned neural field, we use marching cubes [47] and quadric mesh decimation [21] to extract simplified textured meshes $\mathcal{M}$ for the scene. For simplicity, we specify base materials [10] for all the assets and defer material learning to future work. Please see Appendix A.1 for details. Given the desired lighting conditions, we can render our reconstructed scene in a physically-based renderer to model object-light interactions.

## 3.2 Neural Lighting Estimation

In addition to extracting geometry and appearance, we estimate the scene lighting (Fig. 3, left). We use a high-dynamic-range (HDR) panoramic sky dome $\mathbf{E}$ to represent the light from the sun and the sky. This representation well models the major light sources of outdoor daytime scenes and is compatible with rendering engines [7, 30]. Unfortunately, estimating the HDR sky dome from sensor data is challenging, as most cameras on SDVs have limited field-of-view (FoV) and do not capture the full sky. Additionally, camera data are typically stored with low dynamic range (LDR) in self-driving datasets, *i.e.*, intensities are represented with 8-bits. To overcome these challenges, we first leverage multi-camera data and our extracted geometry to estimate an incomplete panorama LDR image that

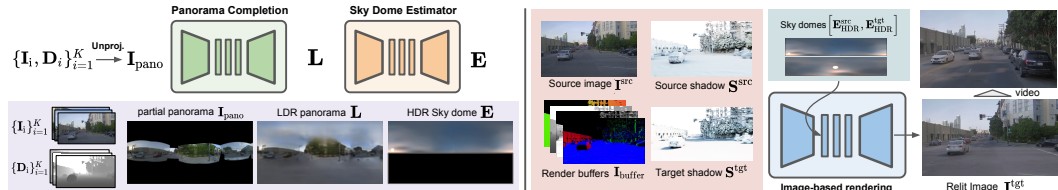

Figure 3: **LightSim modules**. Left: neural lighting estimation. Right: neural deferred rendering.

captures scene context and available sky observations. We then apply an inpainting network to fill in missing sky regions. Then, we utilize a sky dome estimator network that lifts the LDR panorama image to an HDR sky dome and fuses it with GPS data to obtain accurate sun direction and intensity. Unlike prior works that estimate scene lighting from a single limited-FoV LDR image [82, 94, 74], our work leverages multi-sensor data for more accurate estimation. We now describe these steps (as shown in Fig. 3, left) in detail.

**Panorama reconstruction:** Given $K$ images $\mathbf{I} = \{\mathbf{I}_i\}_{i=1}^K$ captured by multiple cameras triggered close in time and their corresponding camera poses $\mathbf{P} = \{\mathbf{P}_i\}_{i=1}^K$, we first render the corresponding depth maps $\mathbf{D} = \{\mathbf{D}_i\}_{i=1}^K$ from extracted geometry $\mathcal{M}$: $\mathbf{D}_i = \psi(\mathcal{M}, \mathbf{P}_i)$, where $\psi$ is the depth rendering function and $\mathbf{P}_i \in \mathbb{R}^{3\times4}$ is the camera projection matrix (a composition of camera intrinsics and extrinsics). We set the depth values for the sky region to infinity. For each camera pixel $(u', v')$, we use the rendered depth and projection matrix to estimate 3D world coordinates, then apply an equirectangular projection $E$ to determine its intensity contribution to panorama pixel $(u, v)$, resulting in $\mathbf{I}_{\text{pano}}$:

$$\mathbf{I}_{\text{pano}} = \Theta\left(\mathbf{I}, \mathbf{D}, \mathbf{P}\right) = E\left(\pi^{-1}(\mathbf{I}, \mathbf{D}, \mathbf{P})\right), \tag{1}$$

where $\Theta$ is the pixel-wise transformation that maps the RGB of limited field-of-view (FoV) images $\mathbf{I}$ at coordinate $(u', v')$ to the $(u, v)$ pixel of the panorama. For areas with overlap, we average all source pixels that are projected to the same panorama $(u, v)$. In the self-driving domain, the stitched panorama $\mathbf{I}_{\text{pano}}$ usually covers a 360° horizontal FoV, but the vertical FoV is limited and cannot fully cover the sky region. Therefore, we leverage an inpainting network [93] to complete $\mathbf{I}_{\text{pano}}$, creating a full-coverage (360° × 180°) panorama image.

**Generating HDR sky domes:** For realistic rendering, an HDR sky dome should have accurate sun placement and intensity, as well as sky appearance. Following [96, 82], we learn an encoder-decoder sky dome estimator network that lifts the incomplete LDR panorama to HDR, while also leveraging GPS and time of day for more accurate sun direction. The encoder first maps the LDR panorama image to a low-dimensional representation to capture the key attributes of the sky dome, including a sky appearance latent $\mathbf{z}_{\text{sky}} \in \mathbb{R}^d$, peak sun intensity $\mathbf{f}_{\text{int}}$, and sun direction $\mathbf{f}_{\text{dir}}$. By explicitly encoding sun intensity and direction, we enable more human-interpretable control of the lighting conditions and more accurate lighting estimation. The decoder network processes this representation and outputs the HDR sky dome $\mathbf{E}$ as follows:

$$\mathbf{E} = \text{HDRdecoder}\left(\mathbf{z}_{\text{sky}}, [\mathbf{f}_{\text{int}}, \mathbf{f}_{\text{dir}}]\right), \text{where } \mathbf{z}_{\text{sky}}, \mathbf{f}_{\text{int}}, \mathbf{f}_{\text{dir}} = \text{LDRencoder}(\mathbf{L}). \tag{2}$$

When GPS and time of day are available, we replace the encoder-estimated direction with the GPS-derived sun direction for more precise sun placement. Please see Appendix A.1 for details.

### 3.3 Learning

We now describe the learning process to extract static scenes and dynamic actor textured meshes, as well as training the inpainting network and sky dome estimator.

**Optimizing neural urban scenes:** We jointly optimize feature grids and MLP headers $\{f_s, f_{\mathbf{k}_d}\}$ to reconstruct the observed sensor data via volume rendering. This includes a photometric loss on the rendered image, a depth loss on the rendered LiDAR point cloud, and a regularizer, as follows: $\mathcal{L}_{\text{scene}} = \mathcal{L}_{\text{rgb}} + \lambda_{\text{lidar}}\mathcal{L}_{\text{lidar}} + \lambda_{\text{reg}}\mathcal{L}_{\text{reg}}$. Specifically, we have

$$\mathcal{L}_{\text{rgb}} = \frac{1}{|\mathcal{R}_{\text{img}}|} \sum_{\mathbf{r} \in \mathcal{R}_{\text{img}}} \left\| C(\mathbf{r}) - \hat{C}(\mathbf{r}) \right\|_2, \ \mathcal{L}_{\text{lidar}} = \frac{1}{|\mathcal{R}_{\text{lidar}}|} \sum_{\mathbf{r} \in \mathcal{R}_{\text{lidar}}} \left\| D(\mathbf{r}) - \hat{D}(\mathbf{r}) \right\|_2. \tag{3}$$

Here, $\mathcal{R}$ represents the set of camera or LiDAR rays. $C(\mathbf{r})$ is the observed color for ray $\mathbf{r}$, and $\hat{C}(\mathbf{r})$ is the predicted color. $D(\mathbf{r})$ is the observed depth for ray $\mathbf{r}$, and $\hat{D}(\mathbf{r})$ is the predicted depth in the range view. To encourage smooth geometry, we also regularize the SDF to satisfy the Eikonal equation and have free space away from the LiDAR observations [89].

**Training panorama inpainting:**   We train a panorama inpainting network to fill the unobserved regions for stitched panorama $\mathbf{I}_{\text{pano}}$. We adopt the DeepFill-v2 [93] network and train on the Holicity [100] dataset, which contains 6k panorama images. During training, we first generate a camera visibility mask using limited-FoV camera intrinsics to generate an incomplete panorama image. The masked panorama is then fed into the network and supervised with the full panorama. Following [93], we use the hinge GAN loss as the objective function for the generator and discriminator.

**Training sky dome estimator:**   We train a sky dome estimator network on collected HDR sky images from HDRMaps [24]. The HDRs are randomly distorted (including random exposure scaling, horizontal rotation, and flipping) and then tone-mapped to form LDR-HDR pairs $(\mathbf{L}, \mathbf{E})$ pairs. Following [82], we apply teacher forcing randomly and employ the $L_1$ angular loss, $L_1$ peak intensity, and $L_2$ HDR reconstruction loss in the log space during training.

# 4   Neural Lighting Simulation of Dynamic Urban Scenes

As is, our lighting-aware digital twin reconstructs the original scenario. Our goal now is to enable controllable camera simulation. To be controllable, the scene representation should not only replicate the original scene but also handle changes in dynamic actor behavior and allow for insertion of synthetic rare objects, such as construction cones, that are challenging to find in real data alone. This enables diverse creation of unseen scenes. As our representation is compositional, we can add and remove actors, modify the locations and trajectories of existing actors, change the SDV position, and perform neural rendering on the modified scene to generate new camera video in a spatially- and temporally-consistent manner. Using our estimated lighting, we can also use a physically-based renderer to seamlessly composite synthetic assets, such as CAD models [76], into the scene in a 3D- and lighting-aware manner. These scene edits result in an "augmented reality" representation $\mathcal{M}', \mathbf{E}^{\text{src}}$ and source image $\mathbf{I}'_{\text{src}}$. We now describe how we perform realistic scene relighting (Fig. 3 right) to generate new relit videos for improving camera-based perception systems.

Given the augmented reality representation $\{\mathcal{M}', \mathbf{E}^{\text{src}}, \mathbf{I}'_{\text{src}}\}$, we can perform physically-based rendering under a novel lighting condition $\mathbf{E}^{\text{tgt}}$ to generate a relit rendered video. The rendered images faithfully capture scene relighting effects, such as changes in shadows or overall scene illumination. However, due to imperfect geometry and noise in material/lighting decomposition, the rendering results lack realism (*e.g.*, they may contain blurriness, unrealistic surface reflections and boundary artifacts). To mitigate this, we propose a photo-realism enhanced neural deferred rendering paradigm. Deferred rendering [16, 65] splits the rendering process into multiple stages (*i.e.*, rendering geometry before lighting, then composing the two). Inspired by recent work [58, 61, 98], we use an image synthesis network that takes the source image and pre-computed buffers of lighting-relevant data generated by the rendering engine to produce the final relit image. We also provide the network the environment maps for enhanced lighting context and formulate a novel paired-data training scheme by leveraging the digital twins to generate synthetic paired images.

**Generate lighting-relevant data with physically-based rendering:**   To perform neural deferred rendering, we place the static background and dynamic actor textured meshes $\mathcal{M}$ in a physically-based renderer [7] and pre-compute the rendering buffers $\mathbf{I}_{\text{buffer}} \in \mathbb{R}^{h \times w \times 8}$, including position, depth, normal and ambient occlusion for each frame. Additionally, given an environment map $\mathbf{E}$ and material maps, the physically-based renderer performs ray-tracing to generate the rendered image $\mathbf{I}_{\text{render}|\mathbf{E}}$. We omit $\mathbf{E}$ in the following for simplicity. To model shadow removal and insertion, we also generate a shadow ratio map $\mathbf{S} = \mathbf{I}_{\text{render}}/\widetilde{\mathbf{I}}_{\text{render}}$, where $\widetilde{\mathbf{I}}_{\text{render}}$ is the rendered image without rendering shadow visibility rays, for both the source and target environment light maps $\mathbf{E}^{\text{src}}, \mathbf{E}^{\text{tgt}}$.

We then use a 2D U-Net [63] that takes the source image $\mathbf{I}^{\text{src}}$, render buffers $\mathbf{I}_{\text{buffer}}$, and shadow ratio maps $\{\mathbf{S}^{\text{src}}, \mathbf{S}^{\text{tgt}}\}$, conditioned on the source and target HDR sky domes $\{\mathbf{E}^{\text{src}}, \mathbf{E}^{\text{tgt}}\}$. This network outputs the rendered image $\mathbf{I}^{\text{tgt}}$ under the target lighting conditions as follows:

$$\mathbf{I}^{\text{tgt}} = \text{RelitNet}\left([\mathbf{I}^{\text{src}}, \mathbf{I}_{\text{buffer}}, \mathbf{S}^{\text{src}}, \mathbf{S}^{\text{tgt}}], [\mathbf{E}^{\text{src}}, \mathbf{E}^{\text{tgt}}]\right). \tag{4}$$

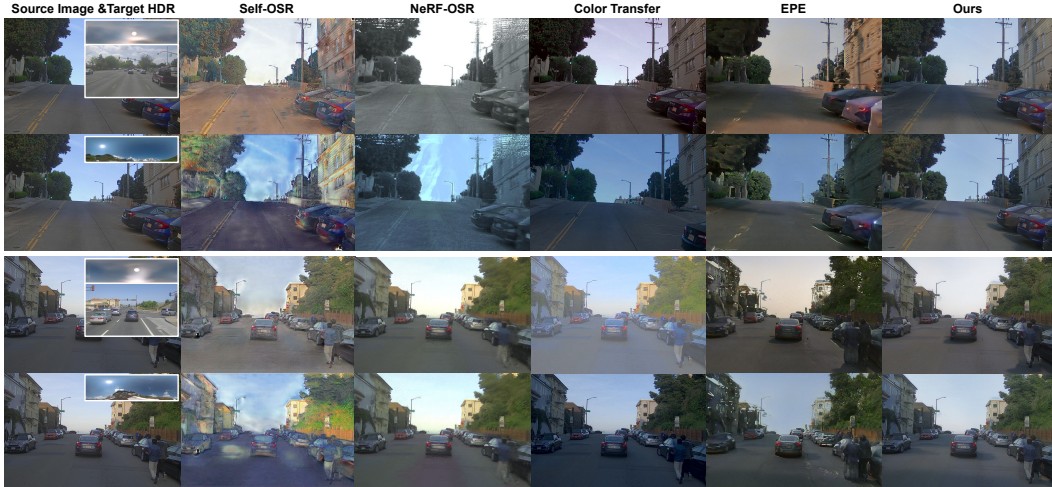

Figure 4: **Qualitative comparison of scene relighting.** For the first and third rows, the real images (other PandaSet snippets) under target lighting conditions are provided for better reference.

This enables us to edit the scene, perform scene relighting, and generate a sequence of images under target lighting as the scene evolves to produce simulated camera videos. The simulation is spatially and temporally consistent since our method is physically-based and grounded by 3D digital twins.

**Learning:** To ensure that our rendering network maintains controllable lighting and is realistic, we train it with a combination of synthetic and real-world data. We take advantage of the fact that our digital twin reconstructions are derived from real-world data, and that our physically-based renderer can generate paired data of different source and target lightings of the same scene. This enables two main data pairs for training the network to learn the relighting task with enhanced realism. For the first data pair, we train our network to map $\mathbf{I}_{\text{render}|\mathbf{E}^{\text{src}}} \to \mathbf{I}_{\text{render}|\mathbf{E}^{\text{tgt}}}$, the physically-based rendered images under the source and target lighting. With the second data pair, we improve realism by training the network to map $\mathbf{I}_{\text{render}|\mathbf{E}^{\text{src}}} \to \mathbf{I}_{\text{real}}$, mapping any relit synthetic scene to its original real world image given its estimated environment map as the target lighting. During training, we also encourage self-consistency by ensuring that, given an input image with identical source and target lighting, the model recovers the original image. The training objective consists of a photometric loss ($\mathcal{L}_{\text{color}}$), a perceptual loss ($\mathcal{L}_{\text{lpips}}$), and an edge-based content-preserving loss ($\mathcal{L}_{\text{edge}}$):

$$\mathcal{L}_{\text{relight}} = \frac{1}{N}\sum_{i=1}^{N}\left(\underbrace{\left\|\mathbf{I}_i^{\text{tgt}} - \hat{\mathbf{I}}_i^{\text{tgt}}\right\|_2}_{\mathcal{L}_{\text{color}}} + \lambda_{\text{lpips}}\underbrace{\sum_{j=1}^{M}\left\|V^j(\mathbf{I}_i^{\text{tgt}}) - V^j(\hat{\mathbf{I}}_i^{\text{tgt}})\right\|_2}_{\mathcal{L}_{\text{lpips}}} + \lambda_{\text{edge}}\underbrace{\left\|\nabla\mathbf{I}_i^{\text{tgt}} - \nabla\hat{\mathbf{I}}_i^{\text{tgt}}\right\|_2}_{\mathcal{L}_{\text{reg}}}\right), \quad (5)$$

where $N$ is the number of training images and $\mathbf{I}^{\text{tgt}}/\hat{\mathbf{I}}^{\text{tgt}}$ are the observed/synthesized label image and predicted image under the target lighting, respectively. $V^j$ denotes the $j$-th layer of a pre-trained VGG network [97], and $\nabla\mathbf{I}$ is the image gradient approximated by Sobel-Feldman operator [70].

## 5   Experiments

We showcase LightSim's capabilities on public self-driving data, which contains a rich collection of sensor data of dynamic urban scenes. We first introduce our experiment setting, then compare LightSim against state-of-the-art (SoTA) scene-relighting methods and ablate our design choices. We then show that our method can generate realistic driving videos with added actors and modified trajectories under diverse lighting conditions. Finally, we show that using LightSim to augment training data can significantly improve 3D object detection.

### 5.1   Experimental Setup

**Datasets:** We evaluate our method primarily on the public real-world driving dataset PandaSet [87], which contains 103 urban scenes captured in San Francisco, each with a duration of 8 seconds

| Method | FID $\downarrow$ | KID ($\times 10^3$) $\downarrow$ |
|---|---|---|
| Self-OSR [94] | 124.8 | $107.1 \pm 4.3$ |
| NeRF-OSR [64] | 143.9 | $94.0 \pm 7.5$ |
| Color Transfer [60] | 85.4 | $29.5 \pm 4.3$ |
| EPE [61] | 93.0 | $56.0 \pm 5.0$ |
| Ours | 87.1 | $30.4 \pm 4.0$ |

Table 1: **Perceptual quality evaluation.**

| Model | mAP (%) |
|---|---|
| Real | 32.1 |
| Real + Color aug. [44] | 33.8 (+1.7) |
| Real + Sim (Self-OSR) | 30.3 (−1.8) |
| Real + Sim (EPE) | 32.5 (+0.4) |
| Real + Sim (Color Transfer) | 35.1 (+3.0) |
| Real + Sim (Ours) | **36.6** (**+4.5**) |

Table 2: **Data augmentation with simulated lighting variations.**

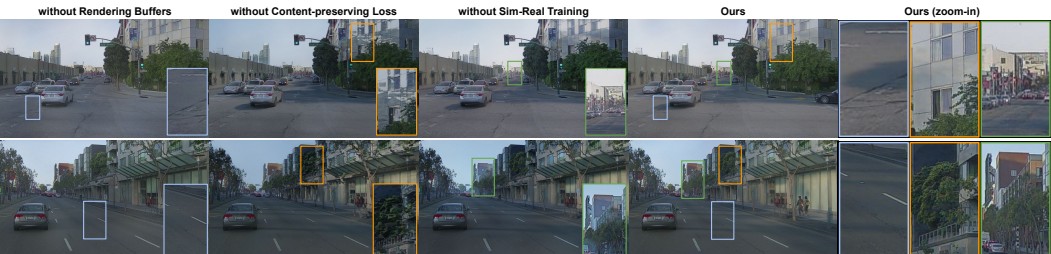

Figure 5: **Ablation study on neural deferred rendering.** Artifacts highlighted with different colors.

(80 frames, sampled at 10hz) acquired by six cameras ($1920 \times 1080$) and a $360°$ 64-beam LiDAR. To showcase generalizability, we also demonstrate our approach on ten dynamic scenes from the nuScenes [11] dataset. These driving datasets are challenging as the urban street scenes are unbounded; large-scale ($> 300\,\mathrm{m} \times 80\,\mathrm{m}$); have complex geometry, materials, lighting, and occlusion; and are captured in a single drive-by pass (forward camera motion).

**Baselines:** We compare our model with several SoTA scene-relighting methods. We consider several inverse-rendering approaches [94, 64], an image-based color-transfer approach [60], and a physics-informed image-synthesis approach [61]. Self-OSR [94] is an image-based inverse-rendering approach that uses generative adversarial networks (GANs) to decompose the image into albedo, normal, shadow, and lighting. NeRF-OSR [64] performs physically-based inverse rendering using neural radiance fields. Color Transfer [60] utilizes histogram-based color matching to harmonize color appearance between images. Enhancing Photorealism Enhancement (EPE) [61] enhances the realism of synthetic images using intermediate rendering buffers and GANs. EPE uses the rendered image $\mathbf{I}_{\mathrm{render}|\mathbf{E}^{\mathrm{tgt}}}$ and G-buffer data generated by our digital twins to predict the relit image.

### 5.2 Neural Lighting Simulation

**Comparison to SoTA:** We report scene relighting results on PandaSet in Table 1. Since the ground truth is unavailable, we use FID [25] and KID [6] to measure the realism and diversity of relit images. For each approach, we evaluate on 1,380 images with 23 lighting variations and report FID/KID scores. 11 of the target lighting variations are estimated from real PandaSet data, while the remaining twelve are outdoor HDRs sourced from HDRMaps [24]. See Appendix C.1 for more details.

Compared to Self-OSR, NeRF-OSR and EPE, LightSim achieves better performance on FID, which indicates that our relit images are more realistic and contain fewer visual artifacts when employed as inputs by ML models. We also show qualitative results in Fig. 4 together with source and target lighting HDRs (Row 1 and 3: relighting with estimated lighting conditions of other PandaSet snippets, Row 2 and 4: third-party HDRs). While Color Transfer achieves the best FID, visually we can see that it only adjusts the global color histogram and does not perform physically-accurate directional lighting (*e.g.*, no newly cast shadows). Self-OSR estimates the source and target lighting as spherical harmonics, but since it must reason about 3D geometry and shadows using only a single image, it produces noticeable artifacts. NeRF-OSR has difficulty with conducting reasonable intrinsic decomposition (*e.g.*, geometry, shadows) and thus cannot perform realistic and accurate scene relighting. EPE incorporates the simulated lighting effects from our digital twins and further enhances realism, but there are obvious artifacts due to blurry texture, broken geometry, and unrealistic hallucinations. In contrast, LightSim produces more reliable and higher-fidelity relighting results

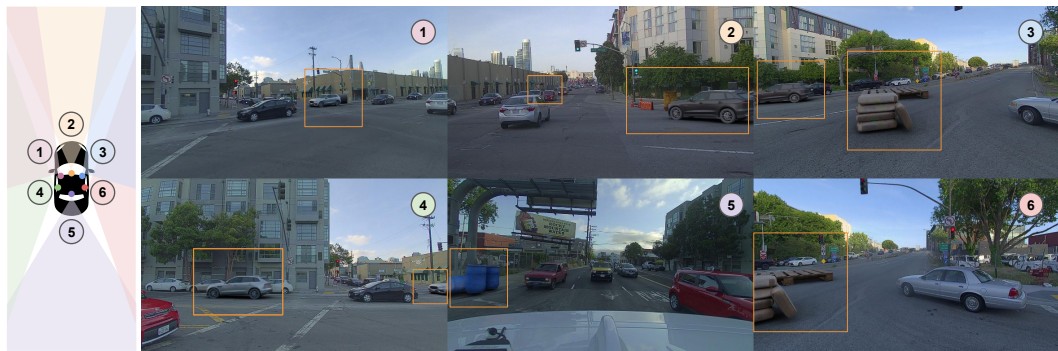

Figure 6: **Lighting-aware multi-camera simulation.** Inserted actors highlighted with boxes.

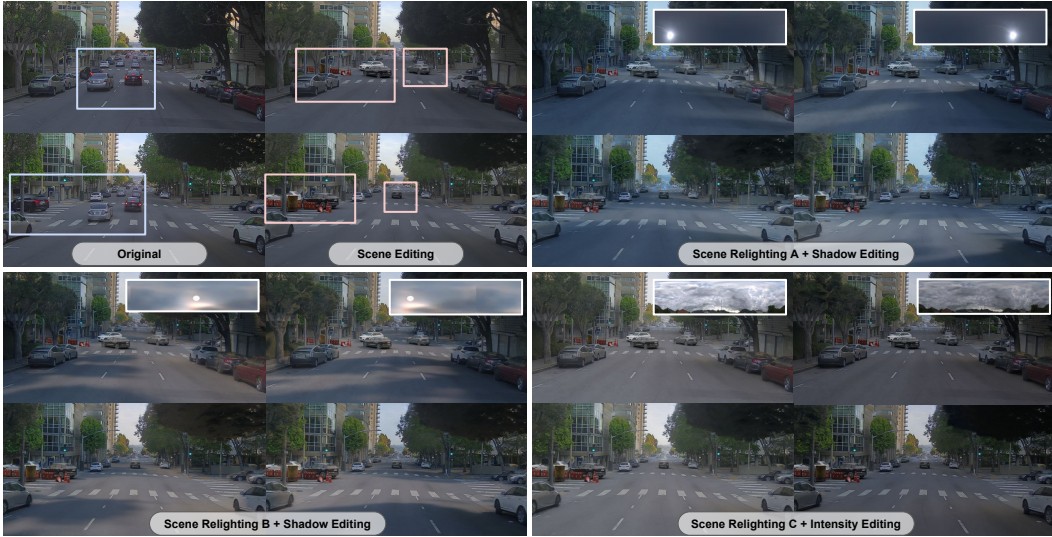

Figure 7: **Lighting-aware camera simulation of novel scenarios.**

under diverse lighting conditions. See Appendix E.1 for more results. In Appendix E.4, we also evaluate LightSim's lighting estimation compared to SoTA and demonstrate improved performance.

**Downstream perception training:** We now investigate if realistic lighting simulation can help improve the performance of downstream perception tasks under unseen lighting conditions. We consider a SoTA camera-based birds-eye-view (BEV) detection model BEVFormer [44]. Specifically, we train on 68 snippets collected in the city and evaluate on 35 snippets in a suburban area, since these two collections are exposed to different lighting conditions. We generate three lighting variations for data augmentation. One lighting condition comes from the estimated sky dome for log-084 (captured along the El Camino Real in California), and the other two are real-world cloudy and sunny HDRs. We omit comparison to NeRF-OSR as its computational cost makes it challenging to render at scale. Table 2 demonstrates that LightSim augmentation yields a significant performance improvement (+4.5 AP) compared to baseline augmentations, which either provide smaller benefits or harm the detection performance.

**Ablation Study:** Fig. 5 showcases the importance of several key components in training our neural deferred rendering module. Pre-computed rendering buffers help the network predict more accurate lighting effects. The edge-based content-preserving loss results in a higher-fidelity rendering that retains fine-grained details from the source image. Training the network to relight synthetic rendered images to the original real image with its estimated lighting enhances the photorealism of the simulated results. Please see more ablations in Appendix E.2.

**Realistic and controllable camera simulation:** LightSim recovers more accurate HDR sky domes compared to prior SoTA works, resulting in more realistic actor insertion (Fig. 6). LightSim inserts the new actors seamlessly and can model lighting effects such as cast shadows for the actors and

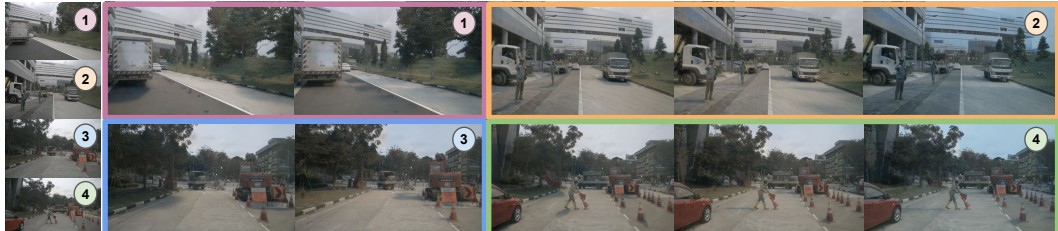

Figure 8: **Scene relighting on nuScenes.** The real images are displayed in the leftmost column.

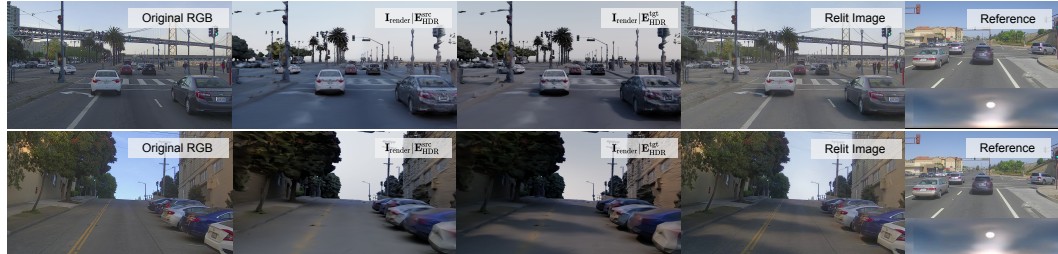

Figure 9: **Physically based rendered images ($\mathbf{I}_{\text{render}}|\mathbf{E}^{\text{src}}$, $\mathbf{I}_{\text{render}}|\mathbf{E}^{\text{tgt}}$) and relighting results.**

static scene, all in a 3D-aware manner for consistency across cameras. Our simulation system also performs realistic, temporally-consistent and lighting-aware scene editing to generate immersive experiences for evaluation. In Fig. 7, we start from the original scenario and perform scene editing by removing all dynamic actors and inserting traffic cones, barriers, and three vehicle actors in the crossroads. Then, we apply scene relighting to change the scene illumination to sunny, cloudy, etc. In Fig. 1, we show another example where we modify the existing real-world data to generate a challenging scenario with two cut-in vehicles.

**Generalization study on nuScenes:**  We now showcase LightSim's ability to generalize to driving scenes in nuScenes [11]. We build lighting-aware digital twins for each scene, then apply a neural deferred rendering model pre-trained on PandaSet. LightSim transfers well and performs scene relighting robustly (see Fig. 8). See Appendix E.6 for more examples.

## 6   Limitations

LightSim assumes several simplifications when building lighting-aware digital twins, including approximate diffuse-only reconstruction, separate lighting prediction, and fixed base materials. This results in imperfect intrinsic decomposition and sim-to-real discrepancies (see Fig. 9). One major failure case we notice is that LightSim cannot seamlessly remove shadows, particularly in bright, sunny conditions where the original images exhibit distinct cast shadows (see Fig. A26). This is because the shadows are mostly baked during neural scene reconstruction, thus producing flawed synthetic data that confuses the neural deferred rendering module. We believe those problems can be addressed through better intrinsic decomposition with priors and joint material/lighting learning [83, 64]. Moreover, LightSim cannot handle nighttime local lighting sources such as street lights, traffic lights and vehicle lights. Finally, faster rendering techniques can be incorporated to enhance LightSim's efficiency [69, 56].

## 7   Conclusion

In this paper, we aimed to build a lighting-aware camera simulation system to improve robot perception. Towards this goal, we presented LightSim, which builds lighting-aware digital twins from real-world data; modifies them to create new scenes with different actor layouts, SDV viewpoints, and lighting conditions; and performs scene relighting to enable diverse, realistic, and controllable camera simulation that produces spatially- and temporally-consistent videos. We demonstrated LightSim's capabilities to generate new scenarios with camera video and leveraged LightSim to significantly improve object detection performance. We plan to further enhance our simulator by incorporating material model decomposition, local light source estimation, and weather simulation.

# Acknowledgement

We sincerely thank the anonymous reviewers for their insightful suggestions. We would like to thank Andrei Bârsan and Joyce Yang for their feedback on the early draft. We also thank the Waabi team for their valuable assistance and support.

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

# Appendix

This appendix details our method, implementation, experimental designs, additional quantitative and qualitative results, limitations, utilized resources, and broader implications. We first detail how we build the lighting-aware digital twins from real-world data (Sec. A.1), then show the network architecture and training details for neural lighting simulation (Sec. A.2). In Sec. B, we provide details on baseline implementations and how we adapt them to our scene-relighting setting. Next, we provide the experimental details for perceptual quality evaluation and downstream detection training in Sec. C. We then report perception quality evaluation at a larger scale with more qualitative comparison with baselines (Sec. E.1) and detailed detection metrics for detection training (Sec. E.3). We demonstrate that our lighting estimation module yields more accurate sun prediction (Sec. E.4) and show additional scene relighting, shadow editing and controllable camera simulation results (Sec. E.5). Finally, we discuss limitations and future work (Sec. F), licenses of assets (Sec. H), computational resources used in this work (Sec. G), and broader impact (Sec. I).

## A  LightSim Implementation Details

### A.1  Building Lighting-Aware Digital Twins of the Real-World

**Neural Scene Reconstruction:**  We first perform neural rendering to reconstruct the driving scene using both front-facing camera images and 360° spinning LiDAR point clouds. We use a modified version of UniSim [89], a neural sensor simulator, to perform neural rendering to learn the asset representations. Unisim [89] incorporates the LiDAR information to perform efficient ray-marching and adopts a signed distance function (SDF) representation to accurately model the scene geometry. To ensure a smooth zero level set, the SDF representation is regularized using Eikonal loss [22] and occupancy loss. To capture the *view-independent* base color $k_d$, we make adjustments to the appearance MLP by taking only grid features as input and omitting the view direction term; all other configurations remain consistent with [89].

After learning the neural representation, we employ Marching Cubes [47] to extract the mesh from the learned SDF volume. Finally, we obtain the vertex base color $k_d$ by querying the appearance head at the vertex location. We adopt base Blender PBR materials [10] for simplicity. Specifically, we use a Principled BSDF with vertex color as the base color. We set materials for dynamic assets and background separately.

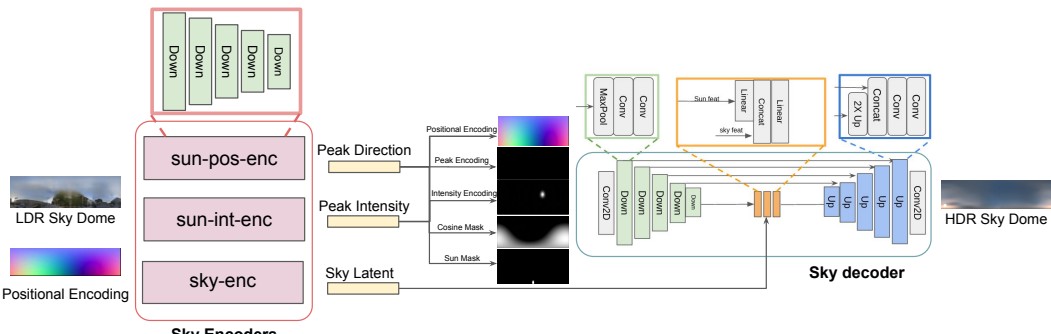

Figure A10: Network architecture of the neural lighting estimation module. Red and blue boxes denote the *sky encoders* and *HDR sky dome decoder* respectively.

**Neural Lighting Estimation:**  We leverage multi-camera data and our extracted geometry to estimate an incomplete panorama LDR image that captures scene context and the available observations of the sky. We then apply an inpainting network to fill in missing sky regions. Finally, we leverage a sky dome estimator network that lifts the LDR panorama image to an HDR sky dome and fuses it with GPS data to obtain accurate sun direction and intensity. To inpaint the panorama image from the stitched multi-camera image, we use the code of the inpainting network DeepFillv2.[1] The network is

---

[1] https://github.com/zhaoyuzhi/deepfillv2

unchanged from its implementation on Github, and it is trained using the hinge loss. Holicity's LDR panoramas are used for training. Each panorama in the training set is first masked using an intrinsics mask – a random mask of observed pixels from a PandaSet log. Distortions are then applied to the mask, including random scaling and the addition of noise. The masked panorama is then fed into the network and supervised with the unedited Holicity panorama.

**Sky dome estimator architecture:** After receiving the full LDR output from the inpainting network, the sky dome estimator network converts the incomplete LDR panorama to HDR. This is achieved by employing GPS and time of day to improve the accuracy of sun direction estimation. To produce an LDR sky dome, only the top half of the panorama is used, as the bottom half does not contribute to global lighting from the sky. The network uses an encoder-decoder architecture, as shown in Fig. A10. The input consists of the LR panorama and the positional encoding map, which is an equirectangular projection that stores a unit vector pointing towards each pixel direction. The positional encoding is concatenated channel-wise with the LDR panorama. Three separate encoders with the same architecture are utilized to estimate the peak direction $\mathbf{f}_{dir}$ and intensity of the sun $\mathbf{f}_{int}$, as well as the latent of the sky $\mathbf{z}_{sky} \in \mathbb{R}^{64}$. To encode the predicted sun peak intensity $\mathbf{f}_{int}$ and peak direction $\mathbf{f}_{dir}$, five intermediate representations with the same size as the input LDR are explicitly encoded. The sky decoder is an U-Net which takes the encoded predicted sun peak intensity $\mathbf{f}_{int}$ and peak direction $\mathbf{f}_{dir}$ and fuses them with the sky latent vector to produce the HDR sky dome.

**Training details:** We train the sky dome estimator using 400 HDRs sourced from HDRMaps [24]. Accurately predicting the peak intensity $\mathbf{f}_{int}$ and direction $\mathbf{f}_{dir}$ is of utmost importance for achieving realistic rendering. Nevertheless, due to the inherently ill-posed nature of this problem, it is challenging to predict these parameters precisely, especially for cloudy skies. Therefore, we propose a dual-encoder architecture, with one encoder trained on HDRs with a clearly visible sun and another trained on all HDR images to capture peak intensity and sky latent more robustly. We empirically find that this works better than using a single encoder trained on all HDR images. The model takes around 12 hours to train with a single RTX A5000.

## A.2   Neural Lighting Simulation of Dynamic Urban Scenes

**Building augmented reality representation:** Given our compositional neural radiance fields, we can remove actors (Fig. 7), add actors (Fig. A23), modify actor trajectories (Fig. 1 and Fig. A23), and perform neural rendering on the modified scene to generate new camera video in a spatially- and temporally-consistent manner. Using our estimated lighting representation, we can also insert synthetic assets such as CAD models [76] into the scenes and composite them with real images in a 3D- and lighting-aware manner (Fig. 1, 7, A23). These scene edits lead to an "augmented reality representation" $\mathcal{M}'$, $\mathbf{E}^{src}$ and source image $\mathbf{I}'_{src}$. We further explain the details of creating augmented reality representations for all the examples we have created in this paper. In Fig. 1, we insert a new CAD vehicle, barriers, and traffic cones and modify the trajectory of the white car. In Fig. 7, we insert three new CAD vehicles (black, white, gray) and construction items and remove all existing dynamic vehicles. In Fig. A23 (top example), we modify the trajectory of the original white vehicle and insert construction items. In Fig. A23 (bottom example), we remove all existing dynamic actors (pedestrians and vehicles), insert three dynamic actors from another sequence 016 with new trajectories, and insert construction items.

**Neural deferred rendering architecture:** The neural deferred rendering architecture is depicted in Fig. A11. It is an image-to-image network adapted from U-Net [63] that consists of four components: *image encoder*, *lighting encoder*, *latent fuser*, and *rendering decoder*. The inputs to the *image encoder* comprise a source RGB image $\mathbf{I}_{src}$, rendering buffers $\mathbf{I}_{buffer}$ (ambient occlusion, normal, depth, and position), and source/target shadow ratio maps $\{\mathbf{S}_{src}, \mathbf{S}_{tgt}\}$; each of these components has 3 channels. In the *latent fuser*, the output of the image encoder is run through a 2D convolution layer, then a linear layer that compresses the input into a latent vector. The image latent and two lighting latent vectors (source and target) are concatenated and upsampled. Finally, the *rendering decoder* upsamples the fused latent vectors and produces the final relit image $\hat{\mathbf{I}}^{tgt} \in \mathbb{R}^{H \times W \times 3}$.

**Learning details:** We train the relighting network using a weighted sum of the photometric loss ($\mathcal{L}_{color}$), perceptual loss ($\mathcal{L}_{lpips}$), and edge-based content preserving loss ($\mathcal{L}_{edge}$), as described in

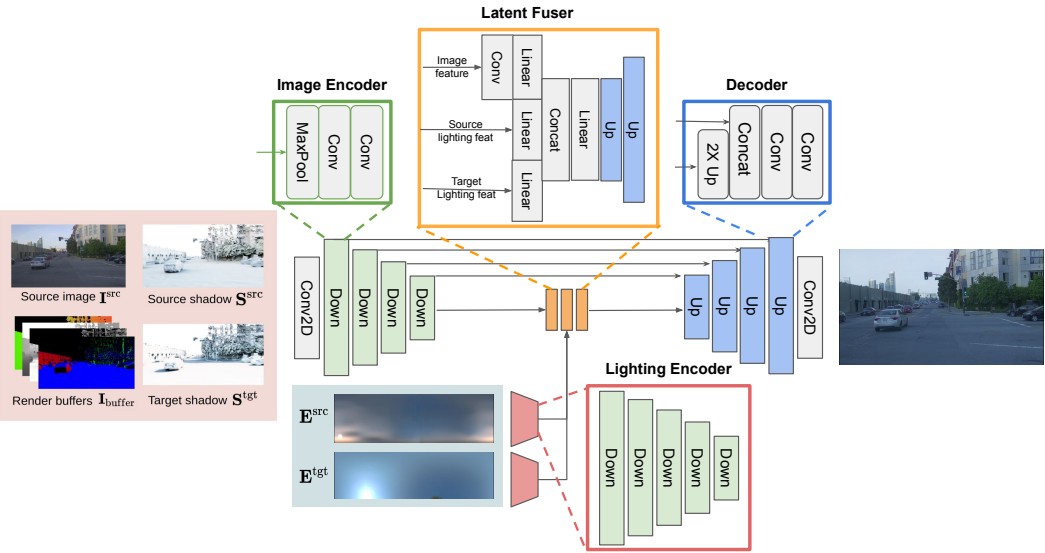

Figure A11: Network architecture of the neural deferred rendering module. Green, red, orange, and blue boxes denote the *image encoder*, *lighting encoder*, *latent fuser* and *rendering decoder*, respectively.

section 4 of the main paper: $\mathcal{L}_{\text{relight}} = \frac{1}{N} \sum_{i=1}^{N} \left( \mathcal{L}_{\text{color}} + \lambda_{\text{lpips}} \mathcal{L}_{\text{lpips}} + \lambda_{\text{edge}} \mathcal{L}_{\text{edge}} \right)$, where $\lambda_{\text{lpips}}$ is set to 1 and $\lambda_{\text{edge}}$ to 400. Our model is trained on pairs of PandaSet [87] scenes, lit with $N_{\text{HDR}} = 20$ HDRs: ten estimated from PandaSet scenes using our neural lighting estimation module, and ten obtained from the HDRMaps dataset [24]. We apply random intensity and offset changes to the HDRs as data augmentation.

# B   Implementation Details for Baselines

For each approach, we evaluate on 1,380 images, applying 23 lighting variations to 15 PandaSet scenes with 4 frames per scene, and report FID/KID scores. To test the diversity and controllability of each method, 11 of the target lighting variations are estimated from real PandaSet data, while the remaining 12 are outdoor HDRs sourced from HDRMaps [24].

## B.1   Self-supervised Outdoor Scene Relighting (Self-OSR)

Self-OSR [94] is an image-based inverse-rendering method that utilizes InverseRenderNet [95] to decompose an image into albedo, normal, shadow, and lighting. Subsequently, it employs two generative adversarial networks (GANs) for neural rendering and sky hallucination, based on the predicted lighting. We use the official pre-trained models[2] and perform inference under novel lighting conditions. If the target lighting originates from PandaSet logs, the model is first applied to a single front-camera image (first frame) to recover the lighting parameters $\in \mathbb{R}^{9 \times 3}$, using an order-2 spherical harmonics model. For outdoor HDRs sourced from HDRMaps, the HDR environment maps are converted into spherical harmonics. The sky masks for PandaSet images are estimated using PSPNet [99].

## B.2   NeRF for Outdoor Scene Relighting (NeRF-OSR)

We use the official code from NeRF-OSR[3] and make several modifications to improve its performance on the PandaSet dataset. Since the self-driving scenes in the PandaSet dataset are usually larger than the front-facing scenarios on the NeRF-OSR dataset, more space needs to be sampled, which presents a challenge. To overcome this, we use LiDAR points as a guide to sample the points

---

[2] https://github.com/YeeU/relightingNet
[3] https://github.com/r00tman/NeRF-OSR

for camera rays by aggregating LiDaR points and creating a surfel mesh. Then, we sample only the points close to the surface $\pm 50cm$ for each camera ray. By doing so, we significantly reduce the number of sampled points to eight in the coarse and fine stages. Without these modifications, NeRF-OSR demonstrates slower convergence rates and struggles to learn reasonable geometry for relighting. Since our inference HDRs (from PandaSet or HDRMaps) differ greatly from NeRF-OSR environment maps (indicating inaccurate scene decomposition by the model), directly applying the environment maps to the scene leads to significant artifacts. Therefore, we retrieve the nearest HDR in the NeRF-OSR dataset during inference for better perceptual quality. For training, we follow the original code base for all other settings. Training one log on NeRF-OSR usually takes 15 hours on 4 T4 GPUs.

### B.3 Color Transfer

Color Transfer [60] uses image statistics in $(L^*, a^*, b^*)$ space to adjust the color appearance between images. We adopt the public Python implementation[4] for our experiments. If the target lighting originates from PandaSet logs, we use the single front-camera image as the target image for color transfer. For outdoor HDRs sourced from HDRMaps, we use the rendered synthetic images (using LightSim digital twins) $\mathbf{I}_{\mathrm{render}|\mathbf{E}^{\mathrm{tgt}}}$ as the target image, as it produces much better performance compared to transferring with the LDR environment map. The latter results in worse performance since the source environment map and target limited-FoV images are in different content spaces.

### B.4 Enhancing Photorealism Enhancement (EPE)

EPE [61] was designed to enhance the realism of synthetic images (*e.g.,* GTA-5 [62] to CityScapes [14]) using intermediate rendering buffers and GANs. We adapt EPE to handle lighting simulation with our established lighting-aware digital twins. Specifically, EPE uses the rendered image $\mathbf{I}_{\mathrm{render}|\mathbf{E}^{\mathrm{tgt}}}$ and rendering buffers $\mathbf{I}_{\mathrm{buffer}}$ generated by our digital twins to predict the relit image. Note that EPE uses the same training data as LightSim, with 20 HDR variations. It also takes all PandaSet front-camera images (103 logs) as the referenced real data. We adopt the official implementation[5] and follow the instructions to compute robust label maps, crop the images, match the crops (5 nearest neighbours) and obtain 459k sim-real pairs. We train the EPE model until convergence for 60M iterations on one single RTX A5000 for around six days.

## C LightSim Experiment Details

### C.1 Perceptual Quality Evaluation

Following [61, 12, 89], we report Fréchet Inception Distance [25] (FID) and Kernel Inception Distance [25] (KID) to measure perceptual quality since ground truth data are not available. Due to NeRF-OSR's large computational cost, we select 15 sequences {001, 002, 011, 021, 023, 024, 027, 028, 029, 030, 032, 033, 035, 040, 053} for quantitative evaluation in the main paper. We also provide Table A3 for larger-scale evaluation (NeRF-OSR excluded), in which 47 sequences (all city logs in PandaSet excluding the night logs and 004 where the SDV is stationary) are used for evaluation. The 47 sequences are {001, 002, 003, 004, 005, 006, 008, 011, 012, 013, 014, 015, 016, 017, 018, 019, 020, 021, 023, 024, 027, 028, 029, 030, 032, 033, 034, 035, 037, 038, 039, 040, 041, 042, 043, 044, 045, 046, 047, 048, 050, 051, 052, 053, 054, 055, 056, 139}.

For each sequence, we select four frames {6, 12, 18, 24} and simulate 23 lighting variations (see Fig. A12). Note that in 23 lighting variations, 20 HDRs (10 estimated HDRs from PandaSet, 10 real HDRs sourced from HDRMaps) are used for data generation to train the LightSim models, while 3 HDRs are unseen and only used during inference. Unless stated otherwise, we use all 8240 real PandaSet images as the reference dataset.

---

[4]https://github.com/jrosebr1/color_transfer
[5]https://github.com/isl-org/PhotorealismEnhancement

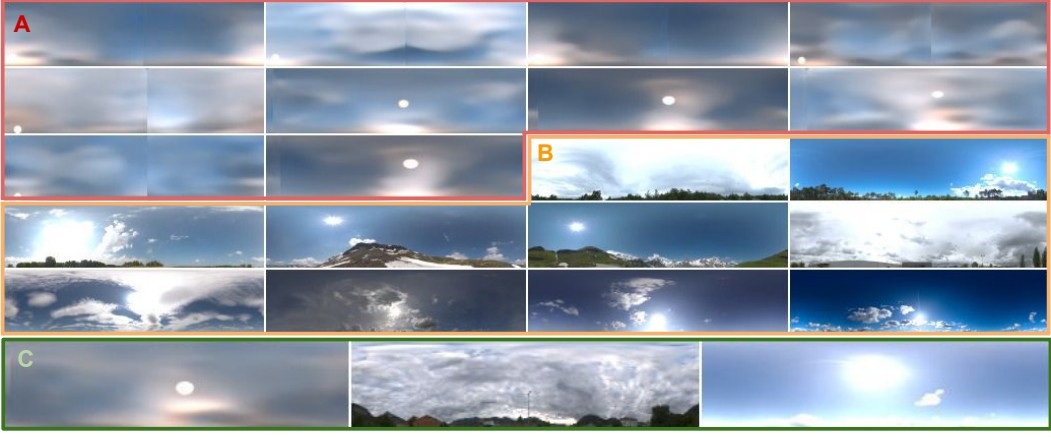

A: 10 training PandaSet HDRs    B: 10 training HDRMaps HDRs    C: 3 unseen inference HDRs

Figure A12: 23 HDR sky domes used for perceptual quality evaluation.

## C.2 Downstream Perception Training

To investigate if realistic lighting simulation can improve the performance of downstream perception tasks under unseen lighting conditions, we conduct experiments on PandaSet using the SoTA camera-based 3D vehicle detection model BEVFormer [44]. We use train on 68 snippets collected in the city and evaluate on 35 snippets in a suburban area, since these two collections are independent and exposed to different lighting conditions. Specifically, the sequences {080, 084, 085, 086, 088, 089, 090, 091, 092, 093, 094, 095, 097, 098, 099, 100, 101, 102, 103, 104, 105, 106, 109, 110, 112, 113, 115, 116, 117, 119, 120, 122, 123, 124, 158} are selected for the validation set, and the remaining sequences are used for training. For the experiments, we use all 80 frames for training and evaluation. We report the average precision (AP) at different IoU thresholds: 0.1, 0.3, and 0.5. The mean average precision (mAP) is calculated as $\text{mAP} = (\text{AP@0.1} + \text{AP@0.3} + \text{AP@0.5})/3.0$.

As shown in Table 2, the integration of LightSim synthetic simulation significantly enhances the performance of monocular detection compared to training with other basic augmentation methods. Further exploration on sufficiently utilizing the simulated data, such as actor behavior simulation or actor insertion, is left to future work.

**BEVFormer Implementation Details:** We use the official repository[6] for training and evaluating our model on PandaSet. We focus on single-frame monocular vehicle detection using the front camera, disregarding actors outside the camera's field of view. The models are trained within vehicle frames using the FLU convention ($x$: forward, $y$: left, $z$: up), with the region of interest defined as $x \in [0, 80\,\text{m}], y \in [-40\,\text{m}, 40\,\text{m}], z \in [-2\,\text{m}, 6\,\text{m}]$. Given memory constraints, we adopt the BEVFormer-small architecture[7] with a batch size of two per GPU. Models were trained for five epochs using the AdamW optimizer [48], coupled with the cosine learning rate schedule.[8] Training each model took approximately six hours on $2\times$ RTX A5000 GPUs. We report the best validation performance across all data augmentation approaches, as models can begin to overfit in the final training stage.

## C.3 Generalization on nuScenes

To evaluate the generalizability of our model, we train the model on PandaSet [87] and evaluate the pre-trained model on nuScenes [11]. The nuScenes [11] dataset contains 1000 driving scenes collected in Boston and Singapore, each with a duration of $\approx 20$ seconds ($\approx 40$ frames, sampled at

---

[6]https://github.com/fundamentalvision/BEVFormer
[7]https://github.com/fundamentalvision/BEVFormer/blob/master/projects/configs/bevformer/bevformer_small.py
[8]https://pytorch.org/docs/stable/generated/torch.optim.lr_scheduler.CosineAnnealingLR.html

2 Hz) acquired by six cameras (Basler acA1600-60gc), one spinning LiDAR (Velodyne HDL32E), and five long-range RADAR (Continental ARS 408-21) sensors. We curate 10 urban scenes from nuScenes [11] characterized by dense traffic and interesting scenarios.

We incorporate front-facing camera and spinning LiDAR and run the neural scene reconstruction module (Section 3.1) to extract the manipulable digital twins for each scene. Then, we utilize the neural lighting estimation module (Section 3.2) to recover the HDR sky domes. This enables us to generate new scenarios and produce the rendering buffers $\mathbf{I}_{\text{buffer}}$ (Eqn. 4 in the main paper) for scene relighting.

## D   Additional Discussions

**Challenges of inverse rendering on urban scenes:**   LightSim assumes several simplifications when building lighting-aware digital twins, including approximate diffuse-only reconstruction, separate lighting prediction, and fixed base materials. Those result in imperfect intrinsic decomposition and sim/real discrepancies. Recent concurrent works such as FEGR [83] and UrbanIR [45] make steps towards better decomposition, but it is still a challenging open problem to recover perfect decomposition of materials and light sources for large urban scenes. As shown in Fig. A13 (top right), the recovered materials bear little semblance to semantic regions in the original scene. These recent relighting works [61, 64, 83, 45] also have shadows baked into the recovered albedo (Fig. A13 left).

We remark that our novelty lies in leveraging neural deferred rendering to overcome the limitations of purely physically-based rendering when the decomposition is imperfect. This allows us to generate better relighting results than prior works that have imperfect decompositions. It is an exciting future direction to incorporate better intrinsic decomposition along with neural deferred rendering for improved relighting.

**Prediction-based vs. optimization-based lighting:**   We explain our design choices in the following two aspects. (a) We use a feed-forward network for lighting estimation, which is more efficient and can benefit from learning on a larger dataset. In contrast, the optimization paradigm is more expensive, requiring per-scene optimization, but has the potential to recover more accurate scene lighting from partial observations. (b) The ill-posed nature of lighting estimation and extreme intensity range make inverse rendering challenging for outdoor scenes [83]. Optimization of the environment map requires a differentiable renderer and high-quality geometry/material to achieve good results. The existing/concurrent state-of-the-art works [64, 83] cannot solve the problem accurately, as shown in Fig. A13 bottom right.

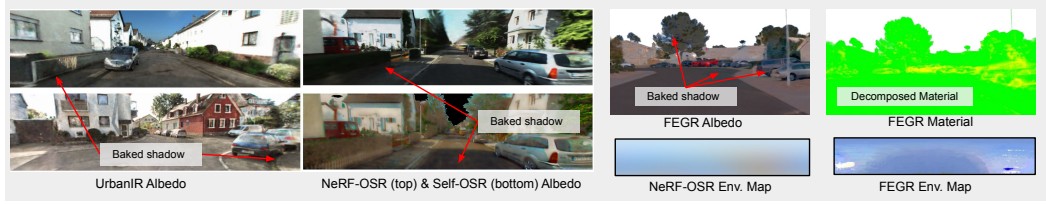

Figure A13: Challenges of inverse rendering on urban scenes for existing works.

**Temporal consistency for neural deferred shading:**   While we do not guarantee temporal consistency, LightSim can produce temporally-consistent lighting simulation videos in most cases. We believe this temporal consistency comes from temporally- and multi-view-consistent inputs during inference (real image, G-buffers), as well as our combination of simulation and real paired relighting data during training. Explicitly enforcing temporal consistency is an interesting direction for future work.

**Random shadowed triangles when relighting due to mesh artifacts:**   We noticed that random shadowed triangles are common in the $\mathbf{I}_{\text{render}|\mathbf{E}^{\text{src}}}$ due to non-smooth extracted meshes. This is more obvious for the nuScenes dataset where the LiDAR is sparser (32-beam) and the capture frequency is lower (2Hz); for this dataset, we notice many holes and random shadowed triangles. However, thanks to our image-based neural deferred rendering pass trained with mixed sim-real data, our relighting network takes the original image and modifies the lighting, which removes many of those artifacts in the final relit images. We show two nuScenes examples in Fig. A14.

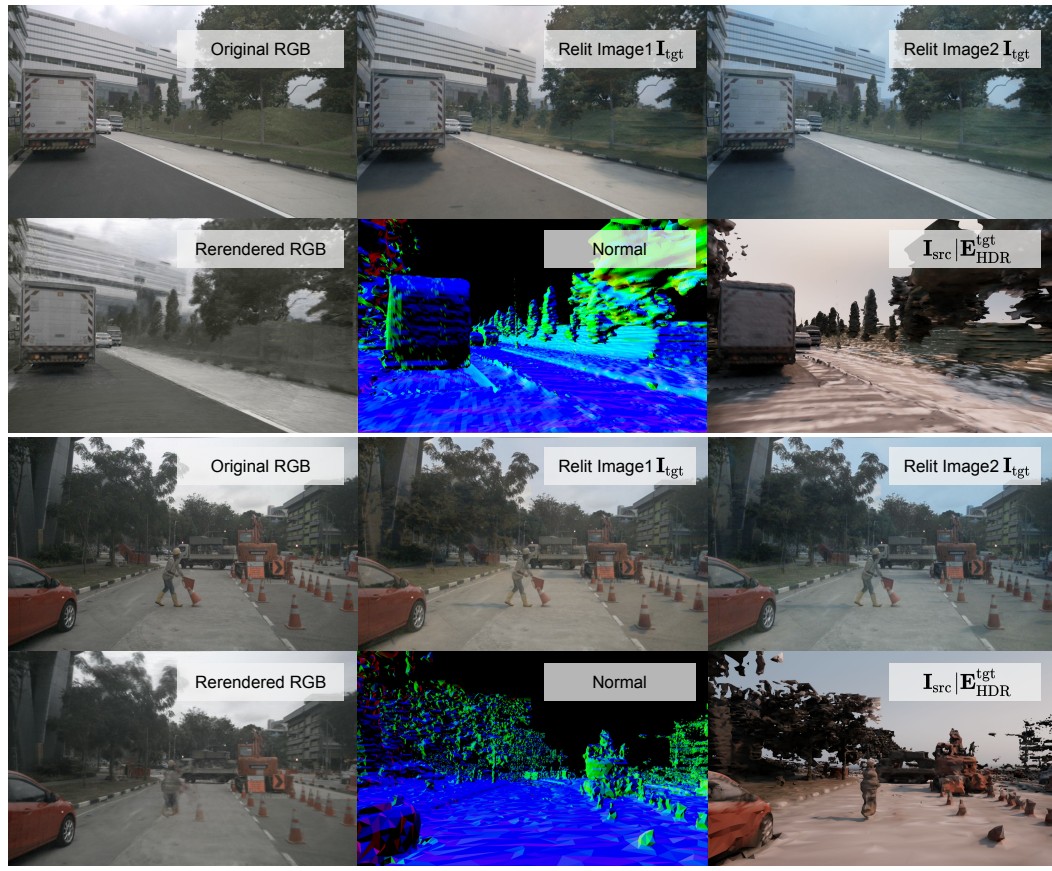

Figure A14: Random shadowed triangles are removed after neural deferred rendering.

# E    Additional Experiments and Analysis

We provide additional results and analysis for scene relighting, ablation studies, downstream training, and lighting estimation. We then showcase more simulation examples using LightSim.

## E.1    Additional Perception Quality Evaluation

Due to the large computational cost of NeRF-OSR [64], we select 15 PandaSet sequences for perceptual quality evaluation in Table 1. Here, we supplement the evaluation at larger scale (47 sequences in total) in Table A3. LightSim achieves perceptual quality (FID and KID) on par with Color Transfer, while the latter approach only adjusts the color histogram and cannot simulate intricate lighting effects properly. Self-OSR and EPE suffer from noticeable artifacts, resulting in significantly worse perception quality and a larger sim-real domain gap.

| Method | FID $\downarrow$ | KID ($\times 10^3$) $\downarrow$ |
|---|---|---|
| Self-OSR [94] | 97.3 | $89.7 \pm 11.5$ |
| Color Transfer [60] | 50.1 | $18.0 \pm 4.5$ |
| EPE [61] | 79.6 | $50.0 \pm 9.1$ |
| Ours | 52.3 | $16.0 \pm 4.6$ |

Table A3: Additional perceptual quality evaluation on 47 sequences.

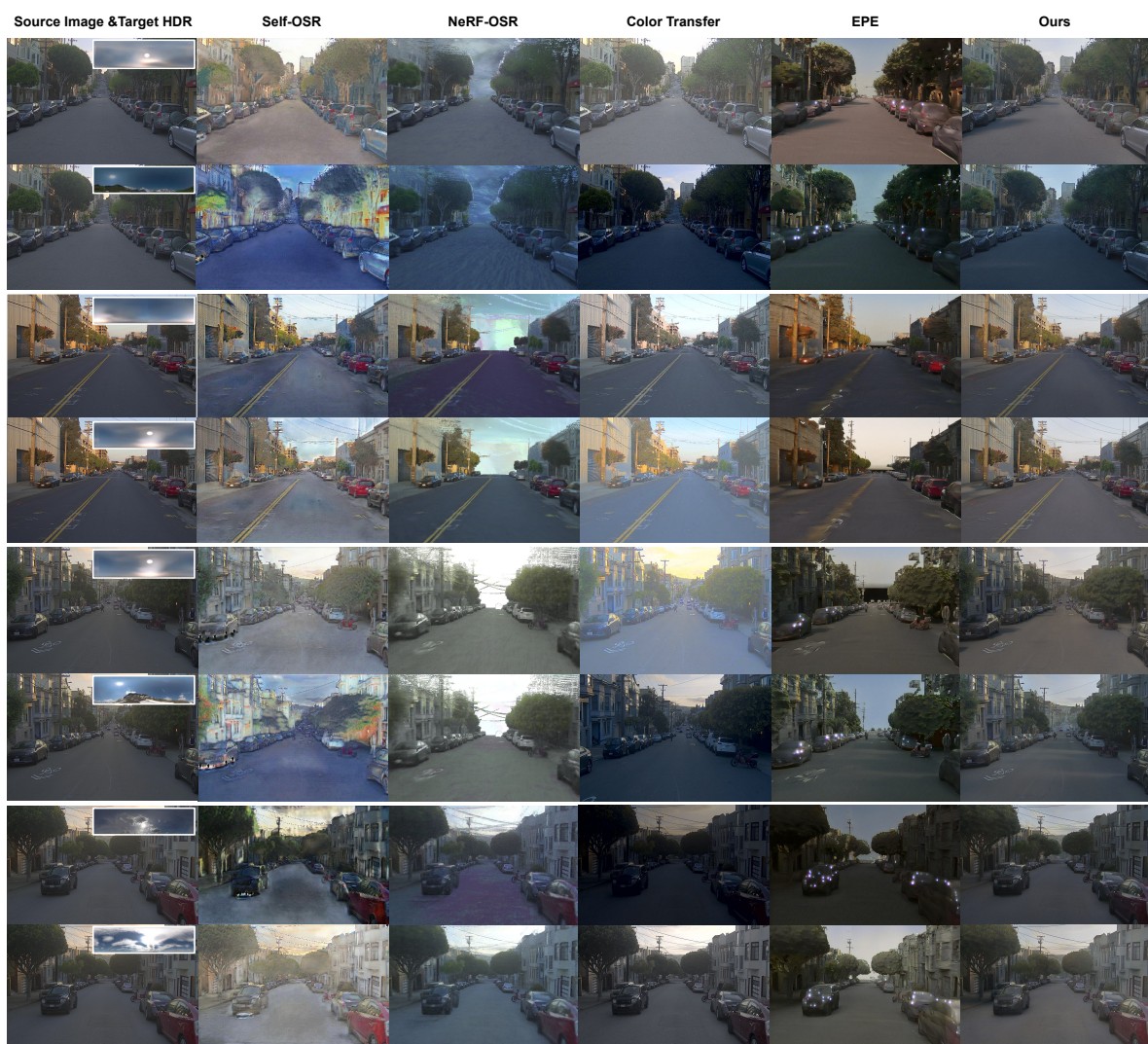

Figure A15: Qualitative comparison against SoTA approaches in scene relighting.

We also provide more qualitative comparisons against SoTA scene relighting approaches in Fig. A15. For each scene, we showcase two different lighting conditions with the inset target HDRs in the leftmost column. We show more scene relighting results of LightSim in Fig. A16 and Fig. A17. We also show results in Fig. A18, where we rotate the HDR skydome and render the shadows at different sun locations, demonstrating controllable outdoor illumination and realistic simulated results. Please refer to the project page for video examples.

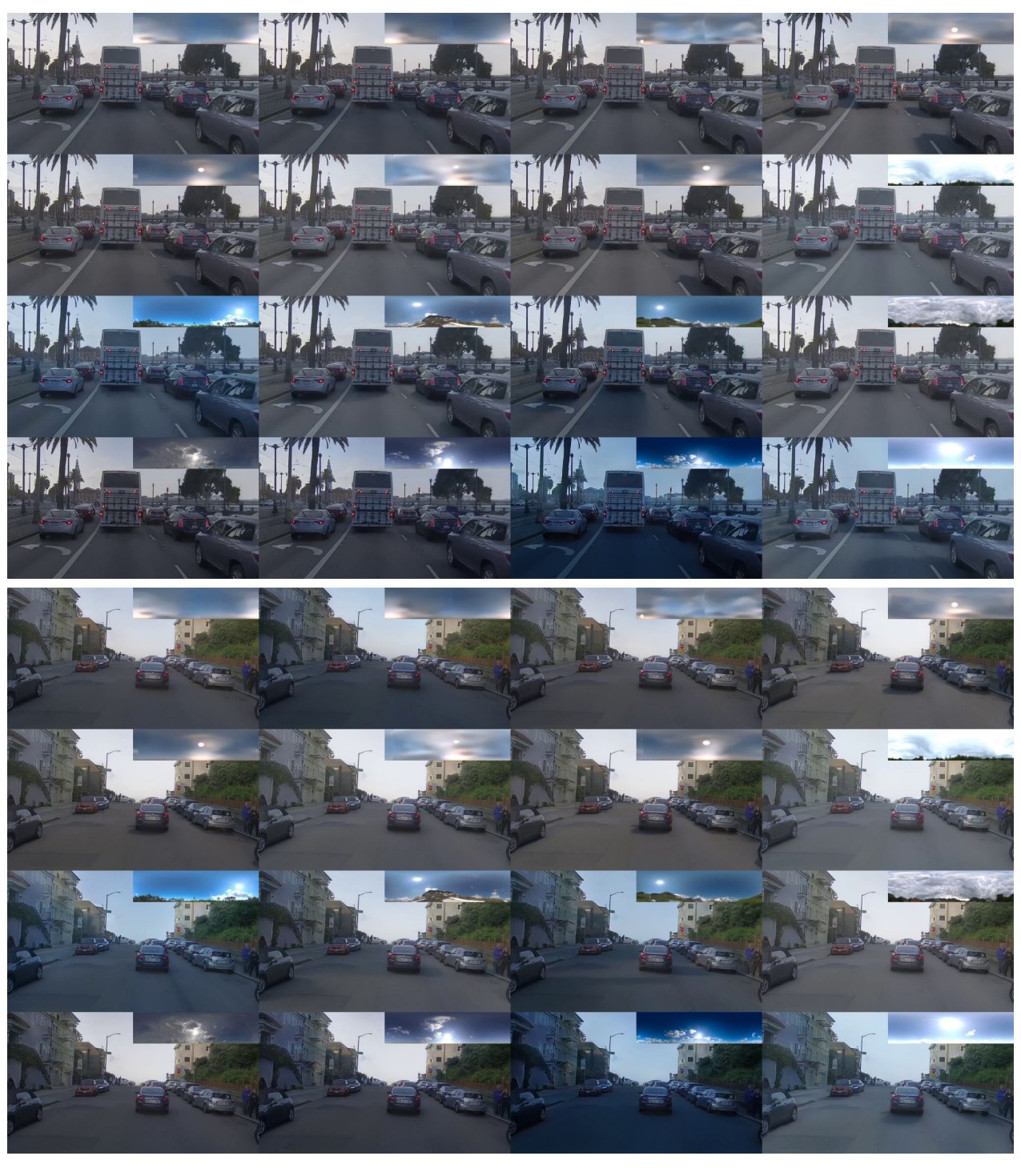

Figure A16: Qualitative examples of scene relighting for LightSim (Part 1).

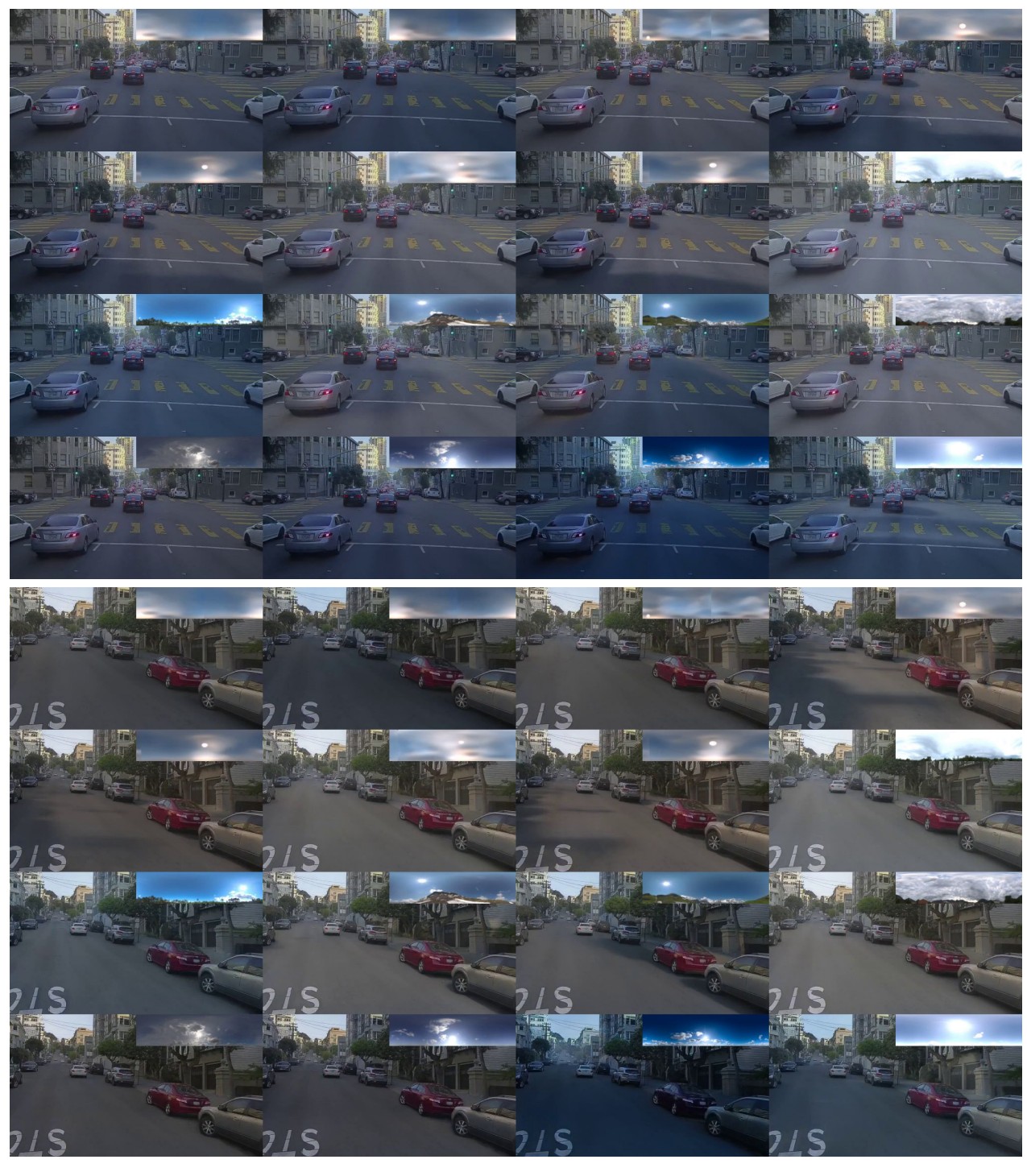

Figure A17: Qualitative examples of scene relighting for LightSim (Part 2).

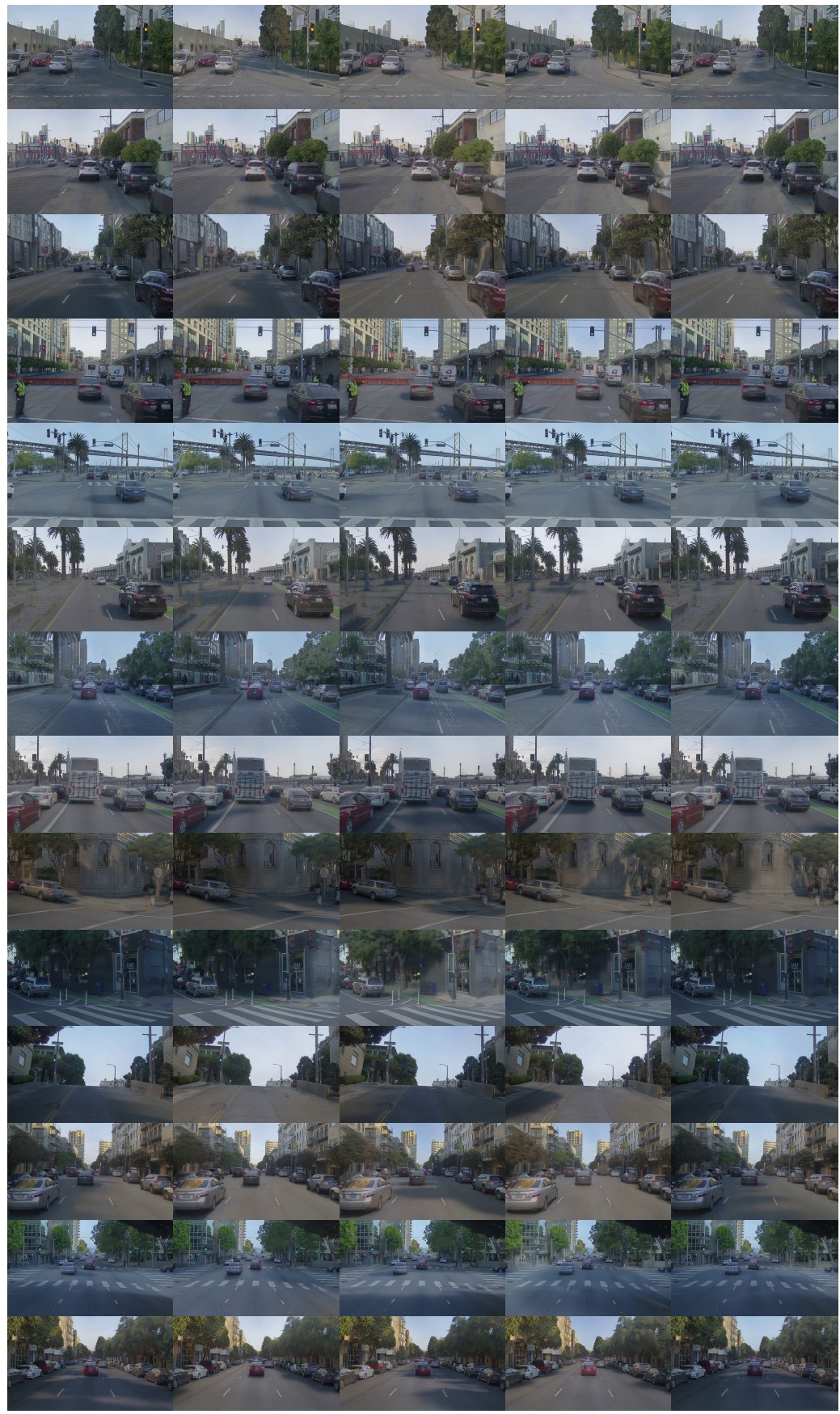

Figure A18: Qualitative examples of scene relighting with shadows edited.

## E.2 Additional Ablation Study

We provide a more thorough ablation study on the important components of the neural deferred rendering module. The perception quality metrics and additional qualitative examples are provided in Table A4 and Fig. A19.

| #ID | Data Pairs sim-real | identity | Rendering Buffers $\mathbf{I}_{buffer}$ | $\{\mathbf{S}^{src}, \mathbf{S}^{tgt}\}$ | Edge loss $\mathcal{L}_{edge}$ | FID ↓ | KID ($\times 10^3$) ↓ |
|---|---|---|---|---|---|---|---|
| 0 | ✗ | | | | | 60.9 | $31.1 \pm 4.0$ |
| 1 | | ✗ | | | | 62.5 | $32.7 \pm 4.0$ |
| 2 | | | ✗ | | | 50.5 | $21.4 \pm 4.1$ |
| 3 | | | | ✗ | | 49.8 | $23.1 \pm 5.0$ |
| 4 | | | | | ✗ | 109.8 | $88.7 \pm 7.3$ |
| 5 | | | | | 200 | 67.1 | $40.9 \pm 4.9$ |
| 6 | | | | | 800 | 57.3 | $31.8 \pm 4.3$ |
| Ours | ✓ | ✓ | ✓ | ✓ | 400 | 55.4 | $27.6 \pm 3.7$ |

Table A4: Ablation studies on LightSim components. For clarity, we only mark the differences between our final model and other configurations. Blank components indicate that the setting is identical to our final model.

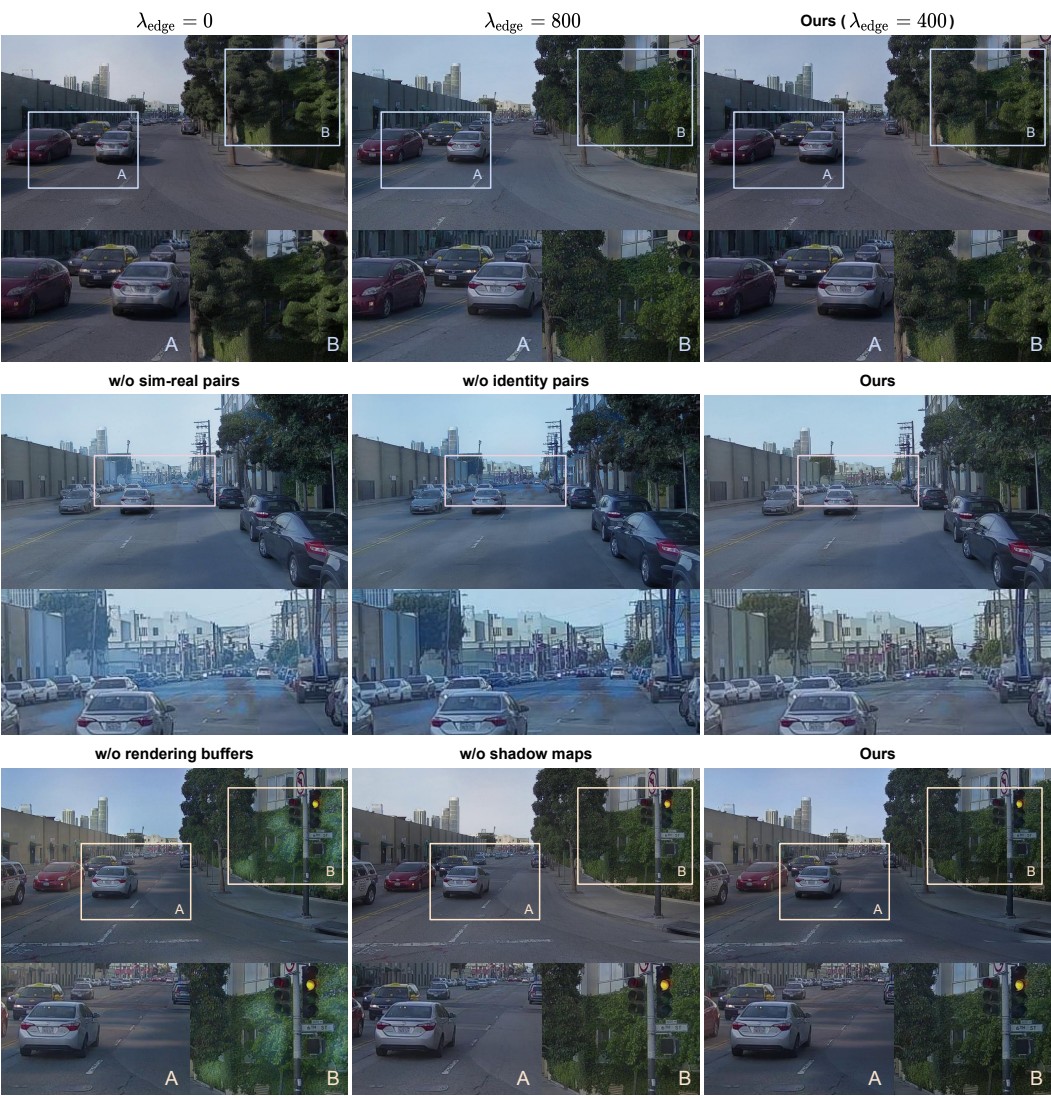

Figure A19: Additional ablation study on neural deferred rendering. LightSim can simulate intricate lighting effects (highlights, shadows) while maintaining realism.

We choose sequence 001 for the quantitative evaluation (FID/KID scores), and all 80 front-camera images are used as the reference dataset. To sufficiently measure rendering quality, we generate $3 \times 80$ relit images using the three unseen HDRs in Fig. A12. We also generate 72 relit images (first frame) with shadows edited using the first unseen HDR. Then, to test generalization to unknown lighting conditions, we pick the first frame and generate 92 relit images using other real HDR maps. Therefore, for each model configuration, there are 404 simulated images used in total for perceptual quality evaluation. We set kid_subset_size=10 for this experiment.

As shown in Table A4 and Fig. A19, *sim-real* and *identity* data pairs provide useful regularization for the neural deferred rendering module by reducing visual artifacts caused by imperfect geometry. Removing those data pairs leads to less realistic simulation results and worse FID/KID scores. On the other hand, the rendering buffers and shadow maps play an important role in realistically simulating intricate lighting effects such as highlights and shadows. We observe unrealistic color and missing cast shadows if pre-computed buffers $\mathbf{I}_{\mathrm{buffer}}$ and shadow maps $\{\mathbf{S}^{\mathrm{src}}, \mathbf{S}^{\mathrm{tgt}}\}$ are removed. Note that the KID/FID metrics are lower when removing rendering buffers since the reference dataset does not include the real data under new lighting conditions; this cannot be interpreted as better visual quality. Finally, we ablate the content-preserving loss $\mathcal{L}_{\mathrm{edge}}$ and find that a proper loss weight helps the model reduce synthetic mesh-like artifacts (compared to $\lambda_{\mathrm{edge}} = 0$) while properly simulating new lighting effects (compared to $\lambda_{\mathrm{edge}} = 800$).

### E.3 Additional Object Detection Metrics

We report detailed detection metrics for perception training with different data augmentation approaches for better reference. Specifically, we report the average precision (AP) at different IoU thresholds: 0.1, 0.3, and 0.5. As shown in Table A5, using LightSim-simulated data yields the best performance improvements. Color Transfer and standard color augmentation [44] are also effective ways to promote the performance of autonomy model under novel lighting conditions. In contrast, Self-OSR and EPE either harm the detection performance or bring marginal gains due to noticeable visual artifacts that cause sim-real domain gap between training and validation.

| Model | mAP (%) | AP@0.1 | AP@0.3 | AP@0.5 |
|---|---|---|---|---|
| Real | 32.1 | 51.2 | 29.5 | 15.7 |
| Real + Color aug. [44] | 33.8 (+1.7) | 53.9 | 31.0 | 16.4 |
| Real + Sim (Self-OSR) | 30.3 (−1.8) | 45.6 | 29.4 | 16.0 |
| Real + Sim (EPE) | 32.5 (+0.4) | 50.2 | 30.6 | 16.7 |
| Real + Sim (Color Transfer) | 35.1 (+3.0) | 55.3 | 32.3 | 17.6 |
| Real + Sim (Ours) | **36.6** (+4.5) | **57.1** | **33.8** | **19.0** |

Table A5: Data augmentation with simulated lighting variations.

### E.4 Comparison with SoTA Lighting Estimation works

We further compare our neural lighting estimation module with the SoTA lighting estimation approaches SOLDNet [74] and NLFE [82]. Table A6 shows the lighting estimation results on PandaSet, where the GPS-calculated sun position is used as reference in error computation. For SOLDNet, we use the official pre-trained model to run inference on limited field-of-view (FoV) front-camera images. For NLFE, we re-implement the sky dome estimation branch without differentiable actor insertion and local volume lighting since the public implementation is unavailable. We also compare a variation of NLFE (named NLFE*) that takes our completed LDR panorama image $\mathbf{L}$ as input. For a fair comparison, LightSim uses the predicted sun position during the encoding procedure. For NLFE and LightSim, the sky dome estimators take the sun intensity and direction explicitly to enable more human-interpretable lighting control. Therefore, we also evaluate the decoding consistency error (log-scale for sun intensity and degree for sun direction). The average metrics are reported on all PandaSet sequences with night logs excluded.

As shown in Table A6, LightSim recovers more accurate HDR sky domes compared to prior SoTA works, with the lowest angular error. It also produces lower decoding error compared to NLFE. Interestingly, we also find that using more camera data (panorama vs limited-FoV image) significantly enhances NLFE's estimation performance and reduces decoding errors. This verifies our idea of leveraging real-world data sufficiently to build the lighting-aware digital twins.

| Method | Input | Angular Error ↓ | Decoding Error ↓ | |
| --- | --- | --- | --- | --- |
| | | | Intensity | Angle |
| SOLDNet [74] | Limited-FoV | 69.98° | – | – |
| NLFE [82] | Limited-FoV | 78.29° | 2.68 | 12.15° |
| NLFE* [82] | Panorama | 47.39° | 2.27 | 8.53° |
| Ours (no GPS) | Panorama | **20.01°** | **1.25** | **1.78°** |

Table A6: Comparisons of sky dome estimation on PandaSet. As reference, our model with GPS leads to 3.78° angular error in sun direction prediction and 1.64° decoding error.

Fig. A20 shows more lighting estimation examples on PandaSet, including stitched partial panorama images $\mathbf{I}_{\mathrm{pano}}$, completed LDR panorama $\mathbf{L}$, and HDR sky domes $\mathbf{E}$. LightSim leverages GPS and time data to get the approximate sun location, enabling recovery of the sun in predicted HDRs even if not observed in partial panorama images $\mathbf{I}_{\mathrm{pano}}$, and the surrounding observed sky and scene context can still be used to approximately estimate the sun intensity.

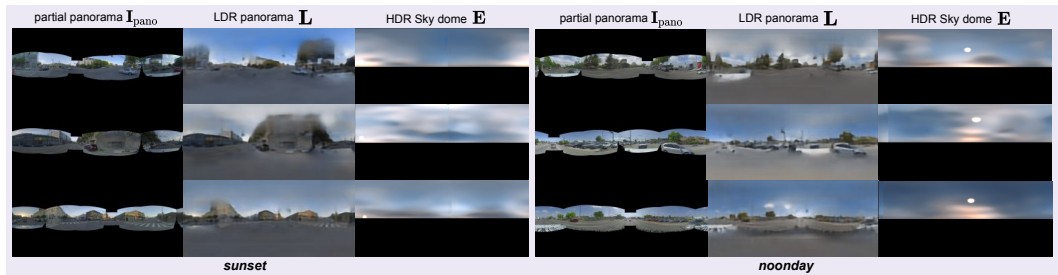

Figure A20: More lighting estimation results on PandaSet.

We further compare virtual actor insertion against SOLDNet and NLFE in Figure A21 on PandaSet sequence 001. We highlight two regions $\{A, B\}$ for comparison to showcase the importance of accurate sun intensity and location prediction, as well as the capability to model inter-object lighting effects. For SOLD-Net and NLFE, we use Poisson surface reconstruction (PSR) [31] to obtain the ground mesh as the plane for virtual actor insertion. Specifically, we first only keep the ground points using semantic segmentation labels, estimate per-point normals from the 200 nearest neighbors within 20cm, and orientate the normals upwards. Then, we conduct PSR with octree depth set as 12 and remove the 2% lowest density vertices.

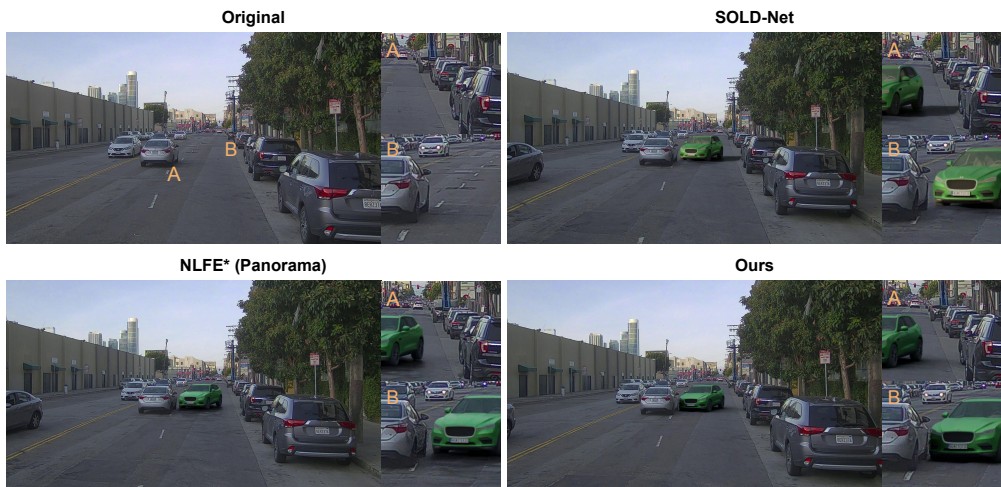

Figure A21: Qualitative comparison of lighting-aware virtual object insertion.

For SOLD-Net, the inserted vehicle looks too bright, with hard shadows cast differently from the other actors since the predicted sun intensity is too strong and the sun direction is not correctly inferred. NLFE estimates the sun intensity and direction more reasonably by consuming our completed LDR panorama image. However, it cannot simulate the shadow cast by the original actor onto the

inserted green vehicle due to the lack of 3D digital twins. In contrast, LightSim can perform virtual actor insertion with inter-object lighting effects simulated accurately thanks to the accurate lighting estimation and 3D world modelling.

In Fig. A22, we show additional lighting-aware actor insertion examples on another PandaSet sequence 024, where the sun is visible in the front camera. LightSim inserts the new actors seamlessly and can model lighting effects such as inter-object shadow effects (between real and virtual objects).

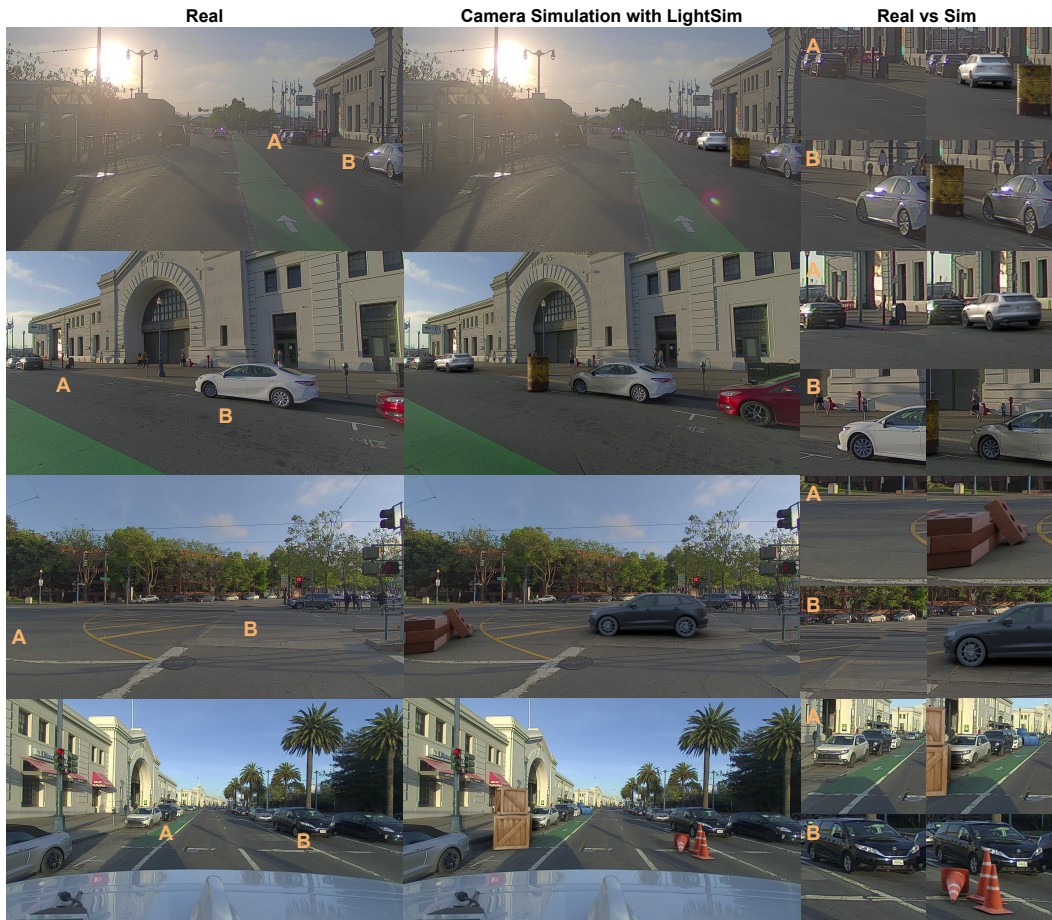

Figure A22: Additional lighting-aware actor insertion examples with LightSim.

### E.5    Additional Camera Simulation Examples

Combining all these capabilities results in a controllable, diverse, and realistic camera simulation with LightSim. In Fig. A23, we show additional camera simulation examples similar to Fig. 1 and Fig. 7 in the main paper. We show the original scenario in the first block. In the second block, we show simulated scene variations with an actor cutting into the SDV's lane, along with inserted traffic barriers, resulting in a completely new scenario with generated video data under multiple lighting conditions. In the third block, we show another example where we add inserted barriers and replace all the scene actors with a completely new set of actors reconstructed from another scene. The actors are seamlessly inserted into the scenario with the new target lighting. Please refer to the project page for video examples.

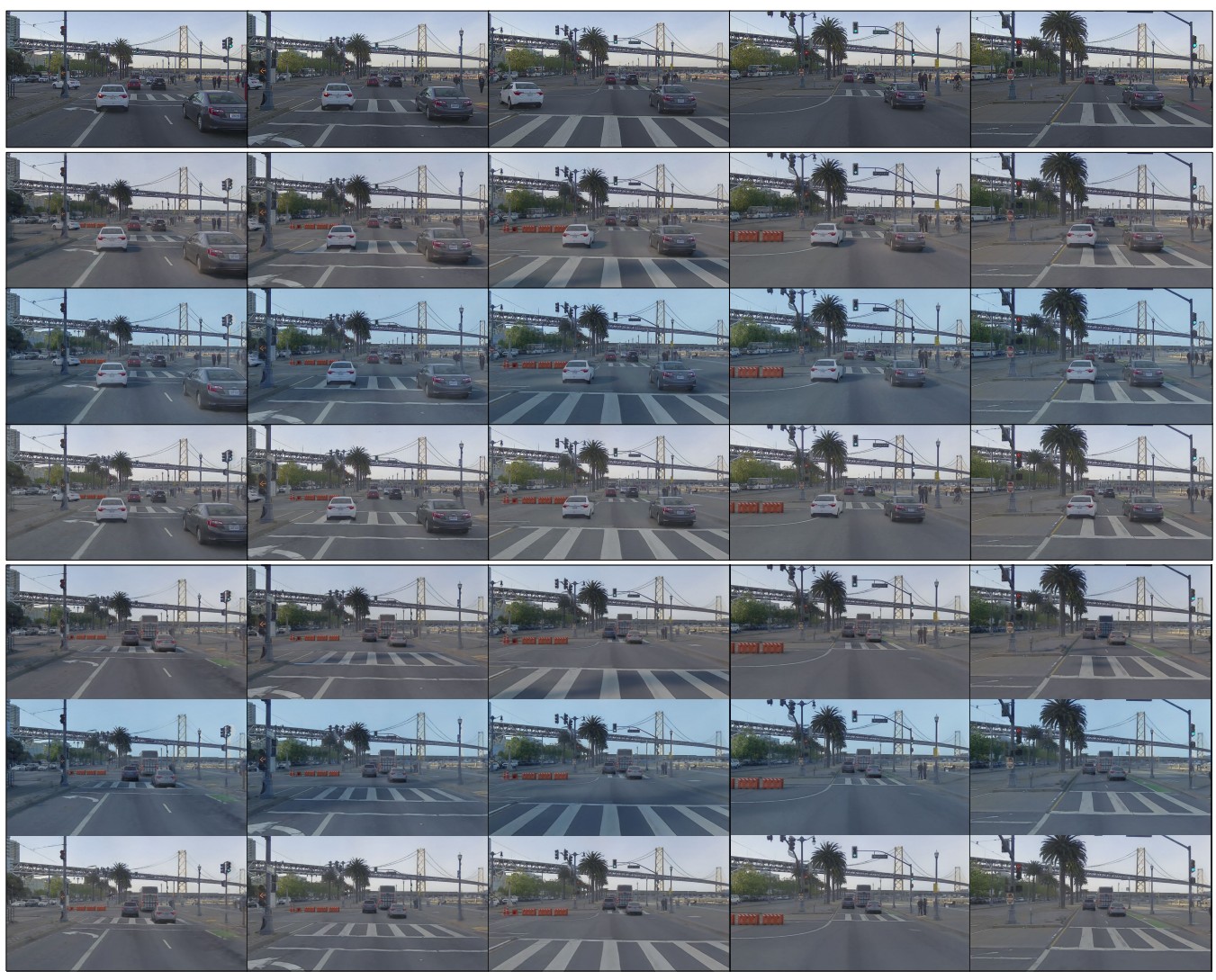

Figure A23: Additional controllable camera simulation examples.

### E.6 Additional Qualitative Results on nuScenes

We further showcase LightSim's ability to generalize to driving scenes in nuScenes. We provide more qualitative scene relighting results in Fig. A24 and Fig. A25. Specifically, we select ten diverse scenarios that involve traffic participants such as vehicles, pedestrians and construction items. The sequence IDs are 011, 135, 154, 158, 159, 273, 274, 355, 544, 763. As described in Sec. C.3, we conduct neural scene reconstruction and lighting estimation (pre-trained on PandaSet) to build the lighting-aware digital twins. Then, we apply the neural deferred rendering model pre-trained on PandaSet to obtain the relit images. Although the nuScenes sensor data are much more sparse compared to PandaSet (32-beam LiDAR, 2Hz sampling rate), LightSim still produces reasonable scene relighting results, indicating good generalization and robustness. Please refer to the project page for video examples.

Occasionally, we observe noticeable black holes (*e.g.,* on log 355 and 763) in the relit images. This is because the reconstructed meshes are low-quality (non-watertight ground, broken geometry) due to sparse LiDAR supervision and mis-calibration. While the neural deferred rendering module is designed to mitigate this issue, it cannot handle large geometry errors perfectly. Stronger smoothness regularization during the neural scene reconstruction step can potentially improve the model's performance.

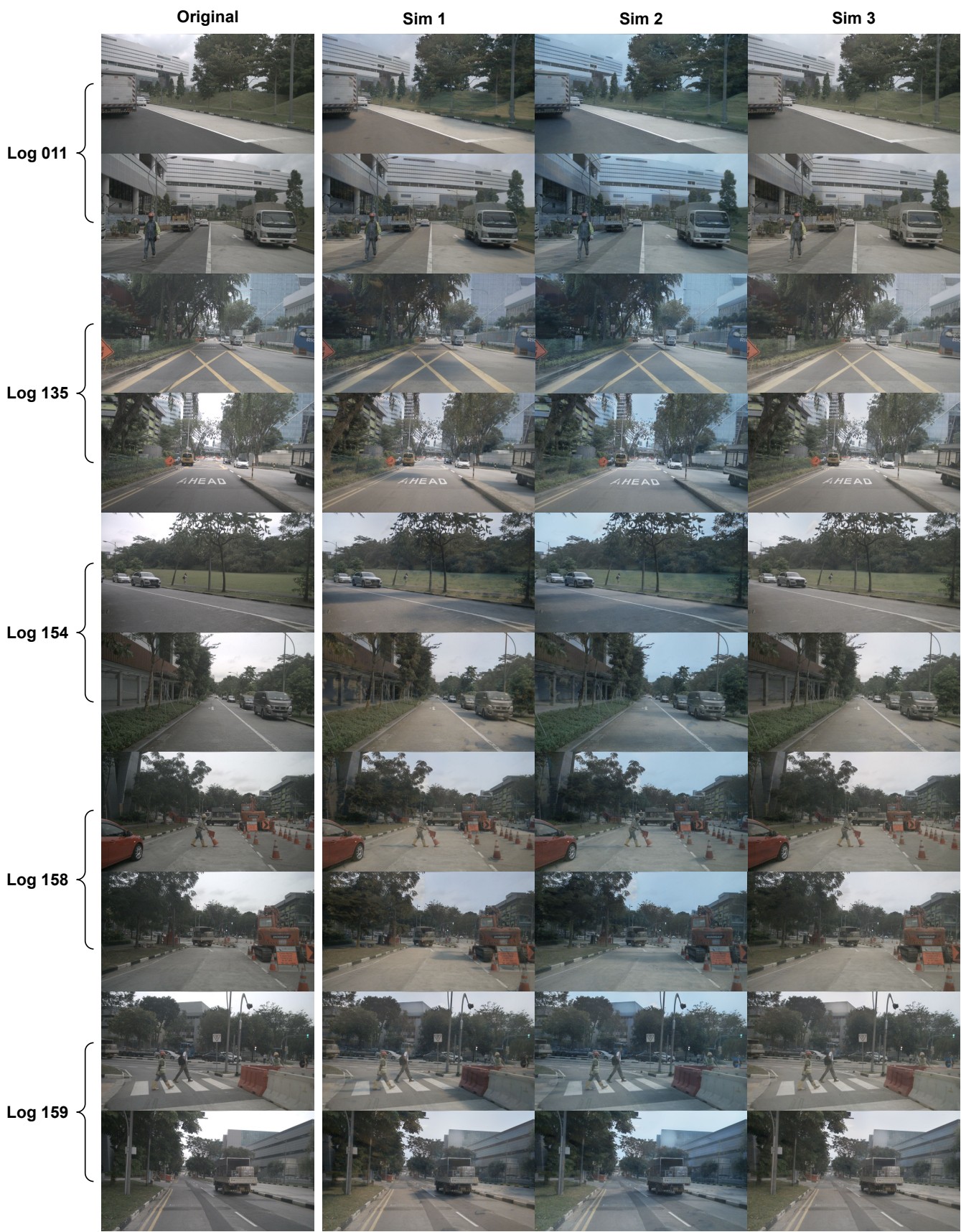

Figure A24: Generalization to nuScenes (Part 1).

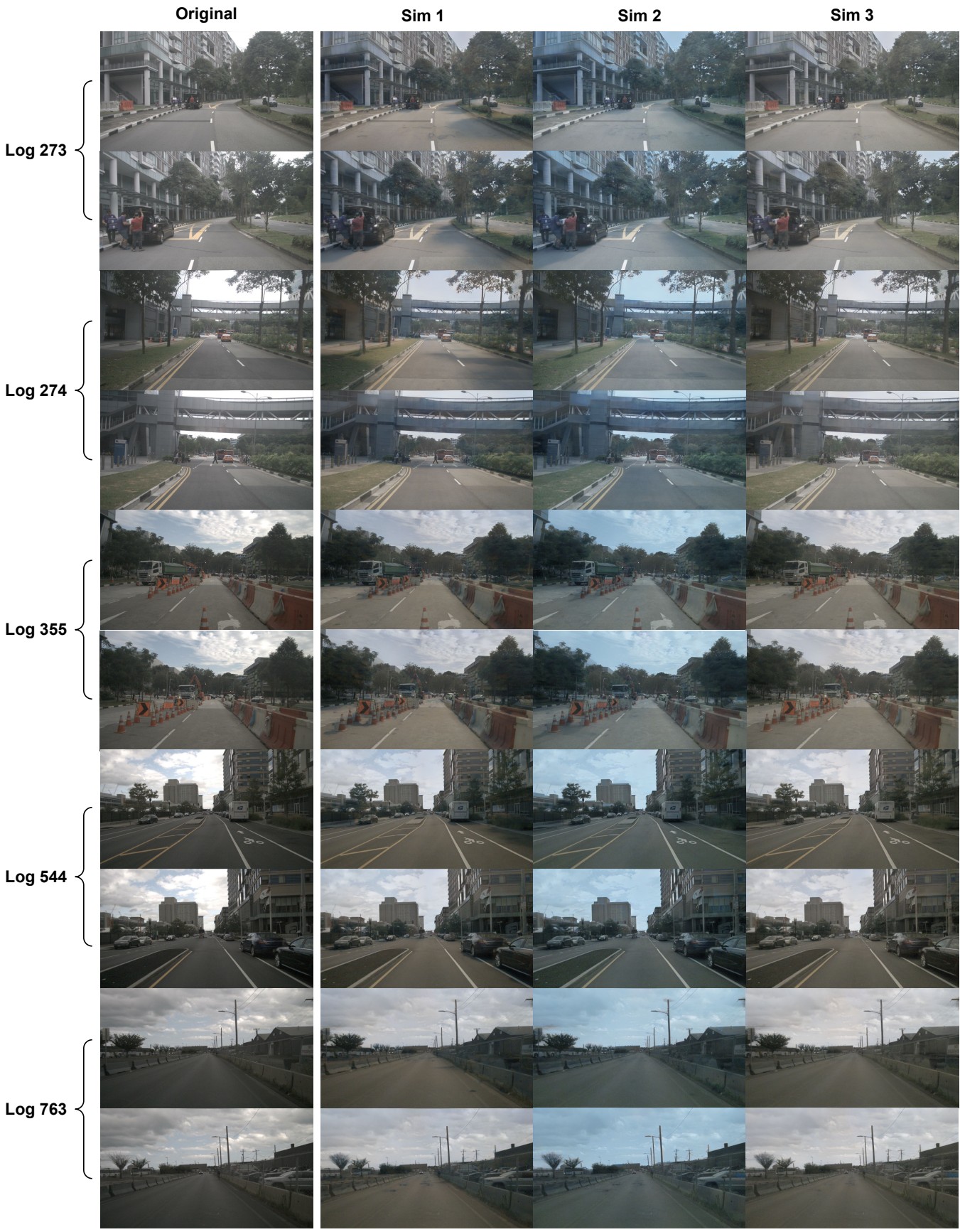

Figure A25: Generalization to nuScenes (Part 2).

# F   Limitations and Future Works

While LightSim can simulate diverse outdoor lighting conditions, there are several areas where it could benefit from further improvements. First, LightSim cannot seamlessly remove shadows, as shown in Fig. A26, particularly in bright, sunny conditions where the original images exhibit distinct cast shadows. This is because the shadows are mostly baked during neural scene reconstruction (see view-independent reconstruction in Fig. A27), producing flawed synthetic data that confuses the neural deferred rendering module. Moreover, we specify fixed materials [10] and predict sky domes that are not ideal for different urban scenes and may cause real-sim discrepancies as shown in Fig. 9. Those issues can potentially be addressed by better intrinsic decomposition with priors and joint material/lighting learning [83, 64]. More discussions for inverse rendering on urban scenes are provided in Sec D.

Second, LightSim uses an HDR sky dome to model the major light sources for outdoor daytime scenes. Therefore, LightSim cannot handle nighttime local lighting sources such as street lights, traffic lights, and vehicle lights. A potential solution is to leverage semantic knowledge and create local lighting sources (*e.g.,* point/area lights [56] or volumetric local lighting [82]). Moreover, our experiments also focus on camera simulation for perception models, and we may investigate the performance of downstream planning tasks in future works. Lastly, our current system implementation relies on the Blender Cycles rendering engine [7], which is slow to render complex lighting effects. Faster rendering techniques can be incorporated to further enhance the efficiency of LightSim [69, 56].

Apart from the further method improvements mentioned above, it is important to collect extensive data from real-world urban scenes under diverse lighting conditions (*e.g.*, repeating the same driving route under varying lighting conditions). Such data collection aids in minimizing the ambiguity inherent in intrinsic decomposition. Moreover, it paves the way for multi-log training with authentic data by providing a larger set of real-real pairs for lighting training.

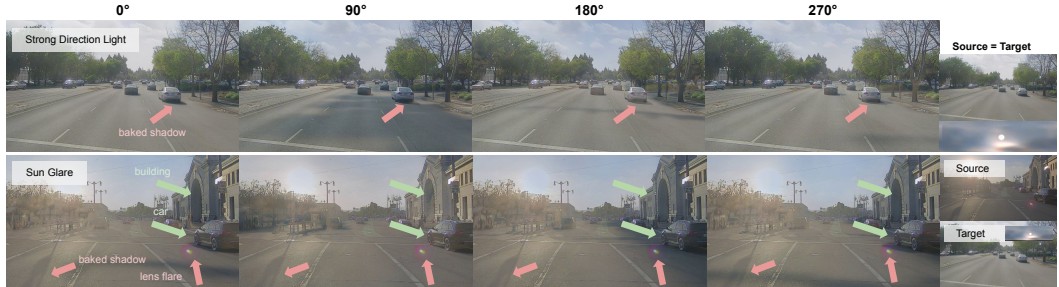

Figure A26: **Failure cases with strong directional lighting.** The neural deferred rendering network cannot fully remove the baked shadows (top row) and other sensor effects (e.g., lens flare – bottom row). We use green and red arrows to highlight areas where LightSim performs well and not well.

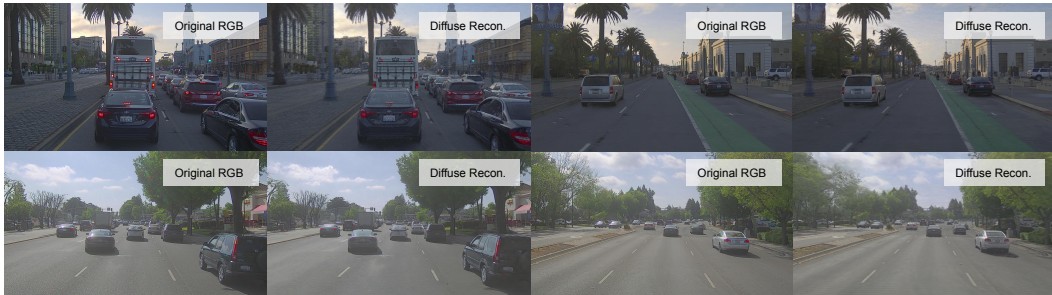

Figure A27: **View-independent reconstruction results for LightSim.** Shadows are baked at this stage, which are then mitigated by neural deferred rendering. The relighting failure cases for the last example are shown in Fig. A26.

In Fig. A26, we highlight two examples of LightSim applied to scenes with strong directional lighting and high sun intensity. Each row shows the shadow editing/relighting results under four different sun angles of the target environment map. In the top row, LightSim cannot fully remove source shadows

in bright and sunny conditions due to the baked shadows in the view-independent reconstruction. Moreover, due to inaccurate HDR peak intensity estimation, the brightness of cast shadows cannot match the original images well. In the bottom row, we depict a source image with high sun intensity and glare relit to a new target lighting. It is challenging to remove the sun glare and alter the over-exposed regions in this setting, but we can still apply some relighting effects to the cars and buildings in the scene (see arrows).

## G  Computation Resources

In this project, we ran the experiments primarily on NVIDIA Tesla T4s provided by Amazon Web Services (AWS). For prototype development and small-scale experiments, we used local workstations with RTX A5000s. Overall, this work used approximately 8,000 GPU hours (a rough estimation based on internal GPU usage reports), of which 3,000 were used for the final experiments and the rest for exploration and concept verification during the early stages of the research project. We provide a rough estimation of GPU hours used for the final experiments in Table A7, where we convert one A5000 hour to two T4 hours approximately.

| Experiment | T4 Hours | Comments |
|---|---|---|
| Table 1 (perceptual quality validation) | 1850 | LightSim (100), NeRF-OSR (1500), EPE (150) |
| Table 2 (downstream training) | 150 | 6× models, each takes 25 GPU hours |
| Table A4 (lighting estimation) | 40 | 15h NLFE, 25h LightSim |
| Fig. 5 & Table A2 (ablations) | 400 | 50h each model |
| Fig. 8 (nuScenes) | 40 | 20h for digital twins, 20h for relighting |
| Others (data generation & demos) | 510 | 500h lighting data generation + 10h demos |

Table A7: Summary of GPU hours used for the final experiments.

## H  Licenses of Assets

We summarize the licenses and terms of use for all assets (datasets, software, code, pre-trained models) in Table A8.

| Assets | License | URL |
|---|---|---|
| Blender 3.5.0 [7] | GNU General Public License (GPL) | https://www.blender.org/ |
| HDRMaps [24] | Royalty-Free[9] | https://hdrmaps.com/ |
| HoliCity [100] | Non-commercial purpose[10] | https://holicity.io/ |
| TurboSquid 3D Models | Royalty-Free[11] | https://www.turbosquid.com[12] |
| PandaSet [87] | CC BY 4.0[13] | https://scale.com/open-av-datasets/pandaset |
| nuScenes [11] | Non-commercial (CC BY-NC-SA 4.0)[14] | https://www.nuscenes.org/ |
| SOLDNet [74] | Apache License 2.0 | https://github.com/ChemJeff/SOLD-Net/ |
| EPE [61] | MIT License | https://github.com/isl-org/PhotorealismEnhancement |
| Self-OSR [61] | Apache License 2.0 | https://github.com/YeeU/relightingNet |
| NeRF-OSR [64] | Non-commercial purpose[15] | https://github.com/r00tman/NeRF-OSR |
| Color Transfer [60] | MIT License | https://github.com/jrosebr1/color_transfer |

Table A8: Summary of the licenses of assets.

## I  Broader Impact

LightSim offers enhancements in camera-based robotic perception, applicable to various domains such as self-driving vehicles. Its ability to generate controllable camera simulation videos (*e.g.*, actor insertion, removal, modification, and rendering from new viewpoints) and adapt to varying outdoor lighting conditions can potentially improve the reliability and safety of intelligent robots for a broad range of environmental conditions. Additionally, LightSim's capacity to create lighting-aware digital twins can improve realism in digital entertainment applications such as augmented reality or virtual reality. However, as with any technology, the responsible use of LightSim is important. Privacy concerns may arise when creating digital twins of real-world locations. We also caution that our system might produce unstable performance or unintended consequences under different datasets, especially when the sensory data are very sparse and noisy.

