# OpenReview forum: "Neural Lighting Simulation for Urban Scenes"
_NeurIPS.cc/2023/Conference — NeurIPS 2023 poster_

### Official Review · Reviewer_G49E · 2023-07-04

**Soundness:** 3 good
**Presentation:** 2 fair
**Contribution:** 2 fair
**Rating:** 4
**Confidence:** 3

**Summary:**

This paper proposed LightSim, a lighting-aware camera simulation system for improving robot perception. This system built relightable digital twins from real-world raw sensor data and enabled applications, e.g., actor insertion, modification, removal, and re-rendering.

**Strengths:**

1. Propose a complete system for outdoor illumination estimation of urban driving scenes and its application.
2. Leverage physics-based rendering to enable controllable simulation of the dynamic scene.
3. Better results than the baseline.

**Weaknesses:**

1. The workload of the paper is large, but it seems an integration of previous works and lacks innovation. e.g., [25] for scene reconstruction, DeepFillv2 for panorama image inpainting, [17] for neural deferred rendering.
2. The motivation for neural deferred rendering needs to be clarified. The motivation for neural deferred rendering needs to be clarified. In the right of Fig. 3, I^src,  E^src, and E^tgt are known, I_buffer, S^src, and S^tgt are generalized by Blender, and digital twins are estimated, then I^tgt can be directly rendered by Blender. Why train an extra network for rendering?
3. What is the material map in L235? Is Blender's default Principle BSDF? and link the vertex color as the base color?
4. Digital Twins is described differently in Fig. 2 and L126-127.

**Questions:**

see the Weakness

**Limitations:**

see the Weakness

---

> ### Author Rebuttal · Authors · 2023-08-09
>
> Thanks for your thoughtful reviews and comments. We address the concerns as follows
>
> **Q1: Novelty - Integration of previous works** \
> **A1:** Our paper’s novelty lies in developing a neural lighting simulation system for self driving, which is critical for thorough evaluation and more robust training of robot autonomy  before safe deployment. To our knowledge, we are one of the first to perform relighting for dynamic urban scenes (SDV and other actors moving in the scenario). LightSim significantly outperforms the state of the art, producing high-quality photorealistic driving videos under a wide range of lighting conditions.
>
> We strongly believe LightSim is a critical and innovative step towards realistic and scalable lighting simulation for robotics. Through this paper, we also hope to convey the importance of leveraging available real data (digital twins) and propose a new regime of exploiting digital twins for lighting simulation.
>
> While some individual components (reconstruction for dynamic scenes [75], environment map in-painting [59] and LDR-HDR lifting [69], G-buffers to aid learning [52]) are studied before, why they are used and how they are used are all carefully designed in the context of self-driving. Specifically, different from existing works that conduct lighting estimation from single image (limited FoV) which is ill-posed and challenging, we fuse all available data (i.e., all six cameras and sun angles based on time and GPS) to reduce ambiguity (see Table A4). To overcome the lack of real world driving scenes captured under different lightings, we propose a “novel data pair training scheme” [Reviewer 6GXP] that leverages the digital twins that are built from real world to generate synthetic paired images under different lighting conditions. Those generated diverse synthetic lighting variation data are then combined with real data to train the neural deferred rendering network. The resulting framework is generic, interpretable and has various capacities including actor insertion, removal, modification, and rendering from new viewpoints, all in lighting-aware manner. We believe LightSim is not just a simple extension or integration of previous works.
> Also, exploiting existing algorithms to realize a novel idea does not mean there is no technical contribution [Reviewer 5Bwf and 6GXP]. We hope the reviewer can acknowledge this. We look forward to follow-up discussions.
>
> **Q2: Clarification of motivation for neural deferred rendering. Why not directly render using Blender?** \
> **A2:** Recovery of perfect geometry, material and lighting in urban driving scenes is a very challenging task which requires strong priors and real-world data under different lighting conditions. To mitigate this issue, we propose to generate synthetic paired lighting variations from our imperfect digital twins and use neural deferred renderer to generate relighting videos in a more realistic manner. Specifically, the neural deferred rendering network learns to relight (guided by coarse blender renderings) while maintaining the realism by taking the original real image as input. The direct blender rendering results have noticeable artifacts as shown in **Rebuttal Figure R3 and R6**.
>
> **Q3: Material map in L234 (default Principle BSDF and link vertex color as base color?)** \
> **A3:** Yes, We use default Principled BSDF in blender and link vertex color as the base color. For all dynamic assets, we set metal=0.5, roughness=0.2, sepcular=0.5, clearcoat = 1.0, clearcoat roughness = 1.0. For the background asset, we set metal=0.0, roughness=0.7, clearcoat = 0.0, specular=0.5. The other material parameters are initialized with Blender default. We will include the details in the revision.
>
> **Q4: “Digital Twins” is described differently in Fig. 2 and L126-127.** \
> **A4:** Both Fig.2 and L126-127 describe lighting-aware digital twins as containing geometry, material and lighting. We look forward to further discussions if additional clarification is required.
>
> We hope the following response addresses the reviewer’s concerns. We look forward to follow-up discussions.

---

> > ### Author Response · Authors · 2023-08-17
> > **Looking forward to follow-up discussions!**
> >
> > We thank the reviewer for taking precious time in checking our responses. We hope our answers and additional results address your concerns well. Specifically,
> >
> > - Q1/A1: We clarified the novelty of LightSim.
> > - Q2/A2: We clarified the motivation of neural deferred rendering and explained why not directly render using Blender (see **Rebuttal Figure R3 and R6**).
> > - Q3/A3: We provided details of material map and will include them in the revision.
> > - Q4/A4: Both Fig.2 and L126-127 describe lighting-aware digital twins as containing geometry, material and lighting.
> >
> > Please let us know if you have any additional or follow-up questions. We will be more than happy to clarify them. Any follow-up discussions are highly appreciated!

---

> > > ### Comment · Reviewer_G49E · 2023-08-18
> > > **Official Comment by Reviewer G49E**
> > >
> > > I am very grateful for the authors' careful answers to my concerns and willing to raise the rate to "borderline accept".
> > >
> > > In addition, the proposed method can estimate SVBRDF and render high-quality relighting results that match the scene. SVBRDF has more applications, such as material editing. Thus, does the neural renderer support more applications, such as material editing?
> > >
> > > If yes, I look forward to displaying some examples in the final version attachment. If not, is it better to use empirical models (such as the Phong model)? After all, PBR is an ill-posed task and is more challenging than empirical models. I look forward to the author explaining or discussing this concern in the next version.

---

> > > > ### Author Response · Authors · 2023-08-19
> > > > **Response to Reviewer G49E**
> > > >
> > > > Thank you for the thoughtful comments and follow-up discussions.
> > > >
> > > > In LightSim, we consider simplified base materials (Q3/A3) and do not conduct material optimization (supp. L343-345). We believe further improvements include incorporating priors to estimate semantic materials of the scene (such as window glass and metal car body)  and stronger regularization [1] during optimization. Exploring material decomposition built on top of LightSim is an exciting future direction. In terms of material editing, LightSim is not designed for this application. We can further investigate if changing the material properties (e.g., making vehicles more reflective by increasing metalness and specular, decreasing the clearcoat) during inference will lead to reasonable material editing results and include results if promising.
> > > >
> > > >
> > > > Both PBR and Phong-based material models could be used for inverse rendering, each with their own strengths and weaknesses. The former is more expressive but might be difficult to optimize, and the latter is easier to optimize but less controllable. It is an interesting direction for future works to investigate joint material decomposition along with geometry and lighting [2, 3].
> > > >
> > > > [1] Neural Fields meet Explicit Geometric Representations for Inverse Rendering of Urban Scenes. Wang et al., CVPR 2023. \
> > > > [2] NeRFactor: Neural Factorization of Shape and Reflectance Under an Unknown Illumination. Zhang et al., Siggraph Asia 2021. \
> > > > [3] Shape, Light, and Material Decomposition from Images using Monte Carlo Rendering and Denoising. Hasselgren et al., NeurIPS 2022.
> > > >
> > > > Thanks again for the prompt and thoughtful reply.
> > > > Please let us know if there are any additional comments or questions.

---

> > > > > ### Author Response · Authors · 2023-08-21
> > > > > **Thanks for your review and post-rebuttal feedback**
> > > > >
> > > > > We thank the reviewers for agreeing to raise the score to borderline accept! Since the discussion period is coming to an end, please let us know if you need anything else from us.

---

### Official Review · Reviewer_6GXP · 2023-07-05

**Soundness:** 4 excellent
**Presentation:** 4 excellent
**Contribution:** 3 good
**Rating:** 7
**Confidence:** 5

**Summary:**

Generating training data for self-driving cars is a challenging task due to the difficulty of capturing real-world scenarios. While video games have been used to generate training data, there exists a domain gap between virtual and real-world environments.
To address these challenges, the authors of this paper propose an approach that generates composable and relightable scenes. The method involves a multi-stage process aimed at training a dynamic Neural Radiance Fields (NeRF) model, which decomposes a scene into static and dynamic components.

Furthermore, the illumination in the scene is learned as a high dynamic range (HDR) sky dome. The proposed approach tackles several sub-problems to achieve its goal. Firstly, panorama reconstruction is performed from the input data. Additionally, in-painting techniques are employed to fill unobserved areas caused by occlusions. An LDR-to-HDR estimation is carried out to enhance the illumination information. Supervision signals are provided by leveraging known sun angles and intensities from GPS data and timestamps. These signals aid in training the model to accurately estimate the scene illumination.

To generate lighting-relevant data, a non-differentiable rendering step is performed, producing essential information such as surface normals, depth, position, and ambient occlusion, along with a render using a single base material.

A 2D U-Net utilizes the render buffer and a target illumination to guide the relighting or editing of a source image. This process allows for consistent relighting of the scene and offers flexibility to insert new objects, as well as move or remove dynamic objects. To train the system, the authors introduce a novel data pair training scheme.

**Strengths:**

- I like the fusing of all available data, especially the supervision from sun angles based on time and GPS data. This is a really strong regularization loss in outside scenes, as the illumination is dominated by sunlight.
- The paired learning scheme for the neural renderer is a good idea and seems to provide good results.
- The editing produces plausible relighting and edits that surpass the quality of purely synthetic data from common game engines.


**Weaknesses:**

- Judging illumination without any specific reference is hard for humans. This is apparent in Fig. 4, where the relighting is hard to judge for plausibility. I suggest taking two images: Source 1 under illumination 1 and Source 2 under illumination 2. Then generate Source 1 under Illumination 2 and vice versa. This way, there exists a reference point for each illumination, and due to the similarity in scenes, it is easier to judge if the illumination is plausible.


**Questions:**

- The geometry from marching cubes often has obvious artifacts, which are especially apparent when lighting the mesh (random shadowed triangles). Have the authors noticed these effects? Is the neural rendering removing them? Is this due to the I_render|E_src -> I_real step?
- The authors propose to only model view-independent diffuse colors instead of the view-dependent typical NeRF one. Did the authors notice any artifacts around reflective surfaces, which degrade the meshes severely?


**Limitations:**

The authors discuss the limitations of their method.

---

> ### Author Rebuttal · Authors · 2023-08-09
>
> Thanks for your thoughtful reviews and comments. We are glad that the reviewer appreciates the novelty and contribution of our work. We address the concerns as follows.
>
> **Q1: Judging illumination is hard for human without reference** \
> **A1:** Thanks for the suggestion. We agree that it is challenging to judge illumination for humans (even experts) without ground truth data or other reference.  In Figure 4, the first and third row are relighting with estimated lighting conditions of other PandaSet log snippets. We added the reference real image in **Rebuttal Figure R5**. We note that the lighting variations of public driving datasets (e.g., PandaSet and NuScenes) are limited, therefore we also use third party HDRs for evaluation (without limited FoV real-image reference). As discussed in Appendix E, it is an important future direction to collect extensive data from real-world urban scenes under diverse lighting conditions (e.g., repeating the same driving route under varying lighting conditions), which would be beneficial to the community (reducing the ambiguity and more helpful for lighting evaluation).
>
>
> **Q2: Random shadowed triangles when relighting due to artifacts in marching cube extracted geometry? (Have the authors noticed these effects? Is neural rendering removing them? Is this due to the $\mathbf{I}_\mathrm{render} | \mathbf{E}^\mathrm{src}$ -> I_real step?)** \
> **A2:** Yes, we noticed that random shadowed triangles are very common in the $\mathbf{I}_\mathrm{render} | \mathbf{E}^\mathrm{src}$ due to non-smooth extracted meshes. This is more obvious for the nuScenes dataset where the LiDAR is sparser (32-beam) and the capture frequency is lower (2Hz). We can notice there are a lot of holes and random shadowed triangles. However, thanks to our image-based neural deferred rendering pass which are trained with mixed sim-real data, our relighting network takes the original image and modifies the lighting, which removes a lot of those artifacts in the final relit images. We show two nuScenes examples in **Rebuttal Figure R6**.
>
> **Q3: Any artifact in the meshes around reflective surfaces due to simple modelling of view-independent diffuse colors.** \
> **A3:** Yes, due to the simplification of view-independent diffuse color and base materials, we cannot accurately simulate the lighting effects around reflective surfaces (e.g., car windows).  Because we leverage both LiDAR and camera data to build the meshes, we did not notice large degradation in the mesh quality for these reflective surfaces.
> We hope LightSim can leverage better intrinsic decomposition in the future to further enhance the performance.

---

> > ### Comment · Reviewer_6GXP · 2023-08-16
> > **Post rebuttal Update**
> >
> > Thanks for the detailed rebuttal and the inclusion of the source illuminations.
> >
> > I have some comments regarding some answers:
> >
> > Q1: There was a paper from Wieschollek et al. - "Learning Robust Video Synchronization without Annotations" which synchronized video of a commute over the duration of a year. I didn't find a direct download link, but maybe for future works, this might be helpful.
> >
> > Q2: My question was more in line with the loss formulation. Which term is responsible for removing these artifacts? It needs to learn to ignore the guidance of the rendered base material image. Is it the  I_render|E_src -> I_real data pair? I think Sim-Real Training in the ablation Fig. 5 of the main paper refers to this supervision data pair.

---

> > > ### Author Response · Authors · 2023-08-16
> > > **Response to Post-rebuttal Update**
> > >
> > > Thanks for the further discussions. We address the comments as follows.
> > >
> > > Q1: Thanks for the suggestion and reference! We believe leveraging ideas from Wieschollek is helpful for future works to synchronize collected multipass lighting data (e.g., same driving route under varying lighting or weather conditions as shown in Wieschollek et al.). The synchronized paired data can be used as evaluation reference and training supervision. We will add more discussions in Appendix E.
> > >
> > > Q2: Yes, to enhance the realism and remove artifacts in simulated data, we train the network to map $\mathbf{I}_{\mathrm{render}|\mathbf{E}^{\mathrm{src}}} \rightarrow \mathbf{I}\_{\mathrm{real}}$, mapping any relit synthetic scene to its original real world image given the estimated environment map. This reduces artifacts (holes and shadowed triangles), encourages the network to be physically grounded, and produces more realistic images (lower FID in Table A.2 and qualitative ablation in Fig. 5).
> > >
> > > Apart from sim-real training, the content-preserving loss $\mathcal{L}_{\mathrm{reg}}$ also plays an important role in reducing the effects of sim artifacts as shown in Fig. 5. This is because we enforce the edges of the relit image to be consistent with the edges of the original real image. This helps the network to ignore/suppress the artifacts in the G-buffers as those artifacts introduce undesired edges. We provided more ablation studies in Fig. A8 and Table A.2 (Supplementary D.2) to better understand the effects of each component in the neural deferred shading network.

---

> > > > ### Author Response · Authors · 2023-08-21
> > > > **Thanks for your review and post-rebuttal feedback**
> > > >
> > > > We thank the reviewers for the follow-up discussions! Since the discussion period is coming to an end, please let us know if you need anything else from us.

---

### Official Review · Reviewer_ENz6 · 2023-07-06

**Soundness:** 3 good
**Presentation:** 2 fair
**Contribution:** 2 fair
**Rating:** 6
**Confidence:** 4

**Summary:**

This paper presents a system for decomposing urban outdoor driving scenes into estimated geometry and lighting components, which are then used as inputs to their deferred neural rendering workflow.  The geometry is represented as a mesh that is extracted using Marching Cubes, from an optimized SDF volume, while the lighting is represented as an inferred HDR sky dome.  A physically based renderer (Blender) is used to render deferred shading passes, which are then provided to a U-Net based "neural renderer" to produce the final output images.  The target application is realistic relighting, to improve diversity of training data for vision-based perception systems in the driving domain.

**Strengths:**

The system presented in this paper involves significant engineering effort, successfully combining multiple learning-based modules.  Their design of the neural deferred renderer module presents some novel extensions beyond prior work, such as the choice of conditioning the U-Net upon the environment map, as well as their loss formulation that includes terms for perceptual loss and edge loss.  Qualitatively, results generally appear to be of a high quality, on par with or marginally better than existing and concurrent works.

**Weaknesses:**

The paper presents a system that draws inspiration from existing works [i.e. FEGR], which makes its own novelty/contribution hard to discern.  It would be helpful for the authors to clarify what the unique contributions of this work are that distinguish it from other related works.

Some terminology is inconsistent and/or using non-standard terms, e.g. use of all of the following terms "physically based rendering", "physics based rendering", and "physics rendering" which all share the same meaning, the latter two of which are not commonly used in existing literature, and therefore may either cause confusion, and in the worst case, be inaccurate.  Another such example is "camera simulation".  I recommend proofreading to improve this particular aspect of the writing.

**Questions:**

In Section 3.1, for the base material that is assigned to reconstructed objects, what are the specific material parameters used?
In Section 3.1, how exactly is the separation between static background and dynamic actors achieved?

On L240, why is it beneficial to use a U-Net to generate the final relit image from the deferred rendering passes?  Given that the deferred rendering passes contain imperfections, does the U-Net

Could you provide more details about the feature grids?  These are mentioned, but not elaborated upon as to their importance, or what purpose they serve.  Do these bear some relation to the multi-resolution hash grids from [Instant-NGP, Müller et al, 2022]?  If so, I suggest including the proper citation, and if not, further elaboration would be ideal.

Could you elaborate on why BEVFormer was chosen for the downstream perception training analysis?

Could you clarify which components of the system are optimized per-scene, and which parts are optimized from a larger dataset?  It seems that the geometry and initial LDR panorama are optimized per-scene, while the other modules are learned from large-scale data -- is this accurate?

**Limitations:**

The relighting results seem to be limited in which aspects of the appearance they affect.  For example, re-rendered shadows look convincing, but the reflections on the cars themselves appear to largely be unaffected in terms of lighting direction / appearance of specular highlights.

Estimation of materials left as future work.

---

> ### Author Rebuttal · Authors · 2023-08-09
>
> Thanks for your thoughtful reviews and comments. We also appreciate the detailed suggestions to improve the presentation quality. We address the concerns as follows.
>
> **Q1: Discern novelty/contribution with FEGR** \
> **A1**: As discussed in the related work section, FEGR is a **concurrent and independent** work (Sec 2 Line 98 to Line 106). It was first made publicly available on arXiv on April 6., **less than two month of NeurIPS submission deadline May 11**. According to NeurIPS policy, “Authors are not expected to compare to work that appeared only a month or two before the deadline.” Therefore, the existence of FEGR should not weaken the novelty of LightSim. Moreover, as stated in related work, while FEGR and LightSim are dealing with a similar task, the methods are different. FEGR aims to conduct inverse rendering (intrinsic decomposition) from a single scene with strong regularization and priors. In contrast, LightSim proposes a “novel data pair training scheme” [Reviewer 6GXP]  and learns on many driven scenes to bypass the challenges of inverse rendering. We believe these two approaches are complementary and the combination of both can lead to a better system. Unlike FEGR, LightSim also produces more realistic relighting videos for **dynamic scenes** (reconstruction and relighting of dynamic actors in the original urban scenes, with inter-object lighting effects) and we also demonstrate the effectiveness of our approach for downstream detection tasks.
>
> We plan to move the discussion in related work into a separate paragraph to make it clearer. Any follow up discussions are highly appreciated.
>
> **Q2: Unify the terminology** \
> **A2:** Thanks for the suggestion. We will unify the terminology (“physically based rendering” and “camera simulation”) to avoid confusion.
>
> **Q3: Base material parameters used for reconstructed objects?** \
> **A3:** Thanks for pointing it out. We use Blender PBR materials. Specifically, we use principled BSDF with vertex color as the base color. For all dynamic assets, we set metal=0.5, roughness=0.2, sepcular=0.5, clearcoat = 1.0, clearcoat roughness = 1.0. For the background asset, we set metal=0.0, roughness=0.7, clearcoat = 0.0, specular=0.5. The other material parameters are initialized with Blender default. We will include the details in the revision.
>
> **Q4: Separation between static background and dynamic actors?** \
> **A4:** We decompose the dynamic scenes into static background and dynamic objects (assuming to be rigid). Specifically, we use 3D bounding box annotations and separate feature grids to model the background and foreground. The neural scene reconstruction is a modified version of [75]. Please refer to Sec A.1 (supp. material) for more details.
>
> **Q5: Why use a U-Net to generate the final relit image from the deferred rendering passes?** \
> **A5:** The U-Net learns to relight from large-scale generated synthetic data under different lighting conditions (low resolution and contain imperfections), meanwhile maintaining the context and quality of original RGB image. By carefully designing the training scheme (sim-to-sim, sim-to-real pair) and learning from large-scale data, our neural deferred rendering pass takes the original RGB image / rendered buffers as inputs and produces high-quality rendering results in a lighting aware manner.
>
> **Q6: Details about feature grids and relation to Instant-NGP, details, citation?** \
> **A6:** We adopt the instant-NGP feature grids with hash encoding. Specifically, we set 16 levels of multi-level feature grids with a hash table size of 2**19. The dimensionality of the feature vector stored in each level's entries is set as 2 and the resolution of the coarsest level is set 16**3. We apologize for missing the citation. We will add the citation and details in the revision.
>
> **Q7: Why was BEVFormer chosen for the downstream perception training analysis?** \
> **A7:** BEVFormer is a state-of-the-art camera only 3D detection model for self-driving scenes, which has been used for comparison and autonomy evaluation in prior works [1, 2, 3, 4].
>
> [1] BEVFusion: Multi-Task Multi-Sensor Fusion with Unified Bird's-Eye View Representation. Liu et al., ICRA 2022. \
> [2] RoboBEV: Towards Robust Bird’s Eye View Perception under Corruptions. Xie et al., ICCV 2023. \
> [3] Benchmarking Robustness of 3D Object Detection to Common Corruptions in Autonomous Driving. Dong et al., 2023. \
> [4] On the Adversarial Robustness of Camera-based 3D Object Detection. Xie et al., 2023.
>
> **Q8: Which components of the system are optimized per-scene, and which parts are optimized from a larger dataset? geometry and initial LDR panorama are optimized per-scene, while the other modules are learned from large-scale data?** \
> **A8:** Yes. The geometry (baked with view-dependent color) and LDR/HDR panorama are optimized / predicted per-scene. The other modules including panorama completion network, LDR2HDR lifting network, neural deferred rendering network are learned from large-scale data.
>
> **Q9: Reflections on the cars themselves appear to largely be unaffected in terms of lighting direction / appearance of specular highlights. Estimation of materials left as future work.** \
> **A9:** We agree that better handling of specular highlights is an exciting research direction that can enhance LightSim. As depicted in Figure R1, this is an open-problem for large-scale scene relighting.
>
> We hope the following response addresses the reviewer’s concern. We look forward to any follow up discussions.

---

> > ### Author Response · Authors · 2023-08-17
> > **Looking forward to follow-up discussions!**
> >
> > We thank the reviewer for taking precious time in checking our responses. We hope our answers and additional results address your concerns well. Specifically,
> >
> > - Q1/A1: We clarified FEGR is a *concurrent and independent* work (less than two months).
> > - Q2/A2: We will unify the terminology.
> > - Q3/A3, Q4/A4, Q6/A6: We explained the technical details and will clarify clearly in the revision.
> > - Q5/A5: We explained why we need neural deferred rendering and the proposed training scheme.
> > - Q7/A7: We justified the choice of BEVFormer.
> > - Q8/A8: We clarified which components are optimized per-scene and which parts are learned from large-scale data.
> > - Q9/A9: We agree better handling of specular highlights is an open, challenging, and exciting research direction (**Rebuttal Figure R1**).
> >
> > Please let us know if you have any additional or follow-up questions. We will be more than happy to clarify them. Any follow-up discussions are highly appreciated!

---

> > > ### Comment · Reviewer_ENz6 · 2023-08-18
> > >
> > > Thank you for answering my questions, I appreciate the time and effort.  And thank you for adding missing citations, and working on improving consistency of terminology.
> > >
> > > Regarding A4: you mentioned "3D bounding box annotations".  How are these obtained?  I assume they are manually annotated?  I personally think it is fine if this is not automatic, as making it automatic would likely be a matter of integrating another component into the system.  However, my opinion is that this should be mentioned clearly in the paper or perhaps in the appendix, to improve the overall reproducibility of your research.
> > >
> > > Aside from these notes, my concerns have been addressed, and I will raise my rating to 'Weak Accept'.

---

> > > > ### Author Response · Authors · 2023-08-18
> > > > **Response to Reviewer ENz6**
> > > >
> > > > Thank you for the thoughtful suggestions and follow-up discussions.
> > > >
> > > > In our experiments, we use the public self-driving datasets (PandaSet and nuScenes) where the human 3D bounding box annotations are provided. We will clarify this clearly in the revision. To further make this automatic, we can integrate off-the-shelf / pretrained models to predict actor tracklets.
> > > >
> > > > Thanks again for the prompt and thoughtful reply. Please let us know if there are any additional comments or questions.

---

> > > > > ### Author Response · Authors · 2023-08-21
> > > > > **Thanks for your review and post-rebuttal feedback**
> > > > >
> > > > > We thank the reviewers for agreeing to raise the score to weak accept! Since the discussion period is coming to an end, please let us know if you need anything else from us.

---

### Official Review · Reviewer_25CV · 2023-07-06

**Soundness:** 3 good
**Presentation:** 3 good
**Contribution:** 3 good
**Rating:** 5
**Confidence:** 4

**Summary:**

The paper proposes a method, LightSim, for recovering geometry, appearance and scene lighting for driving scenes, which enables downstream applications of scene editing and lighting editing. The method incorporates a learned sky dome estimator for hallucinating the original lighting from limited observations, as well as a image-based rendering module for rendering with novel lighting, using rendering proxies as input. The paper demonstrates more realistic light editing results compared to baseline methods in qualitative evaluations, as well as quantitative improvement in scores of perceptual quality and downstream perception tasks where the proposed method is used for training data augmentation.

**Strengths:**

[1] The method is generally novel, and presents reasonable improvements in results. The method takes advantage of full array of sensor data including lidar, RGB, as well as GPS to facilitate reconstruction of scene geometry, lighting including the sun. The paper also introduces a learned rendering model with rendering proxies and lighting cues as input, which is able to alleviate artifacts in rendering.

[2] The paper demonstrates noticeable qualitative improvements when compared to baseline methods. More importantly, given the lack of ground truth images under novel lighting, the paper is able to include indirect quantitative evaluation with perceptual scores and downstream tasks, to demonstrate the method not only produces visually convincing results, but also benefits downstream tasks.

**Weaknesses:**

[1] Clarity. The paper is in general well-written and easy to follow. However, clarify and additional details have to be enhanced for a polished version. For example,

- (L143) details on the representation of base materials for all assets;
- (Sec. 3.2) details on how to acquire geometry for dynamic scenes? Is geometry estimated per-frame? If so how to deal with temporal consistency? If not, how dynamic scene is models in a NeRF-like framework for geometry reconstruction?
- (L246-) What synthetic data is used? What are the specs of the training datasets? What is the training scheme?
- In learning the image-based renderer on real scenes, the estimated sky dome lighting is needed from the previous stage in the pipeline. What if the estimated sky dome lighting is not perfect? Does that affect the learning of the renderer?

[2] Evaluation. The paper is unique in that it leverages a collection of sensor data besides RGB for the task. In this sense, comparing the method to baselines which leverage RGB data only may not be fair. Moreover, is there any reason why the method is not compared against the SOTAs of [69] and [70]? Is it because of the lack of source code and difficulty to reproduce?

**Questions:**

Please see the above section for questions to be addressed. Without understanding those questions, it is difficult to fully evaluate the soundness of the method and results.

**Limitations:**

No potential negative societal impact of the work is discussed.

---

> ### Author Rebuttal · Authors · 2023-08-09
>
> Thanks for your thoughtful reviews and comments. We are glad that our work is recognized as “novel” with “noticeable qualitative improvements”. We address the concerns as follows.
>
> **Q1: Clarifications and additional details** \
> **A1:**  We thank the reviewer for the feedback to improve the presentation quality. We will revise the statements clearly and move more details to the main paper for clarity.
>
> **(1) Base materials for all assets:** \
> We use Blender PBR materials. Specifically, we use principled BSDF with vertex color as the base color. For all dynamic assets, we set metal=0.5, roughness=0.2, sepcular=0.5, clearcoat = 1.0, clearcoat roughness = 1.0. For the background asset, we set metal=0.0, roughness=0.7, clearcoat = 0.0, specular=0.5. The other material parameters are initialized with Blender default. We will include the details in the revision.
>
> **(2) Details on how to acquire geometry for dynamic scenes:** \
> We decompose the dynamic scenes into static background and dynamic objects (assuming to be rigid). Specifically, we use 3D bounding boxes annotation and separate feature grids to model the background and foreground. The neural scene reconstruction is a view-independent version of [75]. Please refer to Sec A.1 (supp. material) for more details.
>
> **(3) Synthetic data used (Line246). What are the specs of the training datasets? What is the training scheme?** \
> We use our reconstructed digital twins to generate synthetic data under different lighting conditions as the paired supervision for neural relighting network training. The training dataset configuration and training scheme are described in detail in the supplementary pdf (Line 110 to 121).
>
> **(4) What if the estimated sky dome lighting is not perfect? Does that affect the learning of the renderer?** \
> If the estimated sky dome is not perfect, the inference and training of the neural deferred rendering network will be affected since the estimated sky dome is taken as the input during inference and we also use estimated lighting to generate synthetic data.
>
>
> **Q2: Comparison to baselines that only leverage RGB information?** \
> **A2:** We tried our best to compare existing works for outdoor relighting and lighting estimation works.  Existing works [1, 2] however. are not designed for self-driving  and do not fully leverage the available data (e.g., LiDAR, time and GPS data). We noticed that [1, 2] are also important baselines compared in concurrent works FEGR [3] and UrbanIR [4].
> Furthermore, we enhance the baseline [5] to leverage the digital twins built by LightSim which leverages all available data as LightSim.
>
> [1] Neural Radiance Fields for Outdoor Scene Relighting. Rudnev et al.., ECCV 2022. \
> [2] Self-supervised Outdoor Scene Relighting. Yu et al., ECCV 2020. \
> [3] Neural Fields meet Explicit Geometric Representations for Inverse Rendering of Urban Scenes. Wang et al., CVPR 2023. \
> [4] UrbanIR: Large-Scale Urban Scene Inverse Rendering from a Single Video. Lin et al., Arxiv 2023. \
> [5] Enhancing Photorealism Enhancement. Richter PAMI, 2021.
>
> **Q3: Comparison with NLFE [69] and FEGR [70]** \
> **A3:** Thanks for the suggestion. We compared with NLFE on the lighting estimation task and provided results in the supplementary material (Table A4 and Figure A9 in Sec D.4 ).  In summary, LightSim achieves more accurate lighting estimation compared to NLFE. LightSim can also model the inter-object lighting effects compared to NLFE in actor insertion application.
>
> For FEGR, we notice it is a *concurrent and independent* work (Sec 2 Line 98 to Line 106). It was first made publicly available on arXiv on April 6., **less than two month from the  NeurIPS submission deadline May 11**. According to NeurIPS policy, “Authors are not expected to compare to work that appeared only a month or two before the deadline.” Additionally, since FEGR does not offer its source code, reproducing it within a tight deadline becomes challenging.
>
> We hope the following response addresses the reviewer’s concern. We look forward to any follow up discussions.

---

> > ### Author Response · Authors · 2023-08-17
> > **Looking forward to follow-up discussions!**
> >
> > We thank the reviewer for taking precious time in checking our responses. We hope our answers and additional results address your concerns well. Specifically,
> >
> > - Q1/A1: We clarified the technical details.
> > - Q2/A2: We tried the best to compare existing works for outdoor relighting and lighting estimation. We enhance EPE to leverage all available data as LightSim.
> > - Q3/A3: We compared with NLFE (Table A4 and Figure A9 in Sec D.4). For FEGR, we notice it is a *concurrent and independent* work (less than two months).
> >
> > Please let us know if you have any additional or follow-up questions. We will be more than happy to clarify them. Any follow-up discussions are highly appreciated!

---

> > > ### Comment · Reviewer_25CV · 2023-08-19
> > >
> > > I would like to thank the authors for the clarifications in the rebuttal. Indeed the rebuttal solved many of my questions, by referring to the supplementary material (which I initially entirely missed out). I would recommend in a later version of the main paper, adding more pointers to the supplementary about important ablations such as the where my initial questions were raised. I also understand that limited baselines were available by the time of submission and thus comparison not needed. Nevertheless it would be great that In case the authors get a chance to add comparison in a later version, they should do so for a more thorough and contemporary evaluation.
> > >
> > > One more question to be discussed it the dependency of the later stage on the estimated lighting. The authors should add results to demonstrate this potential issue (e.g. failure cases as a result of wrong estimated lighting). However this is an acceptable limitation common to methods of long learning-based pipelines, where errors propagate from early stage to later ones. To sum up, I would encourage the authors to add additional results and statement/analysis on this issue.
> > >
> > > Overall thanks again to the rebuttal and I will keep my original rating favoring acceptance.

---

> > > > ### Author Response · Authors · 2023-08-20
> > > > **Response to Reviewer 25CV**
> > > >
> > > > Thank you for the thoughtful suggestions. We are glad that your concerns are well addressed. We will add more pointers to the supplementary and include additional discussions in the next version.
> > > >
> > > > **Examples of relighting failures due to imperfect lighting estimation**: Thanks for the suggestion. Since the estimated sky domes are taken as the inputs, imperfect lighting estimation may lead to inaccurate relighting results. For example, if the peak direction and intensity for source/target environment map have large deviations from the ground truth, the brightness and casted shadows for relit images will be affected. We will provide additional qualitative examples and analysis in the revision.
> > > >
> > > > Thanks again for the prompt and thoughtful reply. Please let us know if there are any additional comments or questions.

---

> > > > > ### Author Response · Authors · 2023-08-21
> > > > > **Thanks for your review and post-rebuttal feedback**
> > > > >
> > > > > We thank the reviewers for the follow-up discussions! Since the discussion period is coming to an end, please let us know if you need anything else from us.

---

### Official Review · Reviewer_5Bwf · 2023-07-06

**Soundness:** 3 good
**Presentation:** 4 excellent
**Contribution:** 2 fair
**Rating:** 5
**Confidence:** 3

**Summary:**

This work extends a recent novel view synthesis approach for autonomous driving scenes with relighting and virtual object insertion with properly cast shadows. The proposed method is a system that consists of the following steps: i) geometry and albedo reconstruction from sensor data, ii) estimation of the environment map (inpainting the sensor data and lifting LDR to HDR), iii) neural deferred rendering that takes in the source image and relights it based on a new environment map. The proposed system enables relighting the original source images or novel views rendered from the step (i). Apart from this, it also allows virtual object insertion of either reconstructed or synthetic assets. The proposed approach is thoroughly evaluated on the tasks of relighting, and virtual object insertion on Pandas and Kitti datasets. Furthermore, it is used to simulate data for a downstream task (3D object detection) where it boosts the performance when compared to using only real world data (a subset of all scenes) . In most experiments/evaluation metrics, the proposed outperforms the selected baselines.

**Strengths:**

- This work is a well-designed system that efficiently builds upon prior work (UniSIM). While individual components are not very special, their combination is technically sound and leads to good results.
- The experimental evaluation is really thorough with several ablation studies and qualitative/quantitative results including the evaluations on the downstream tasks.
- I agree with the authors that trying to recover perfect materials and geometry in AV scenes is a very challenging task that requires strong priors. Therefore, I really like the use of the neural deferred renderer.
- The paper is well written and easy to follow, the motivation for the approach is very clear, and the design choices are well-supported by the results/ablation studies.

**Weaknesses:**

From my perception, the following are the most important weaknesses of this work:

- **Simplification of the reconstruction process**: During the reconstruction stage ((i) above) two strong simplification are made which in my opinion prevent obtaining good results on challenging illuminations (strong directional light). First, (If I understood this part correctly) the method aims to reconstruct the albedo of the scene, by simply removing the directional dependence of the color MLP, but the supervision still comes from the full RGB images. Second, a single material (no information which?) is used for all the assets after the reconstruction. However, material differences in AV scenes are quite large (e.g. asphalt compared to metallic cars). While the reconstructed albedo images are not shown in the submission (if they are, and I missed them, I am sorry), but I assume that in case of strong directional light the shadows simply get baked into the "albedo" representation.

- **Temporal aspect**: The prediction of the relighted frames is done "independently" for each frame using a deferred neural renderer. This means that the temporal aspect is not considered and there is no guaranty for mutliview consistency.

- **Recovering the env map**: The env map is recovered in two steps, i) the sensor data is projected onto a panorama and inpainted, ii) the LDR image is lifted to HDR. However, if the sun is not observed in the sensor data, I assume that it is very difficult for the inpainting network to predict its location (this is actually very ill-posed as set up). While GPS information can help with the sun location, it cannot predict the occlusion, by clouds. I think that integrating the optimization of the ENV map in the first step and guide the sun location/intensity with the shadows in the scene would be more principled.

- **Somewhat limited novelty**:  This is actually not a major weakness in my mind, but still wanted to bring it up. The individual components of the proposed system are in my mind not very novel (e.g. reconstruction is from UniSim, env in-painting and lifting to HDR has often been done before).  This being said, I do think that the whole system and the combination of these modules are sufficiently novel, but probably  not be too far away from the border of acceptable/expected novelty.

**Questions:**

**Comments**:
L7: I would suggest softening the claim that the reconstructed assets are *"relightable digital twins*". The two simplifications mentioned above are in contradiction with this statement.

I think that tit would be very beneficial to show some results of the UniSIM albedo reconstruction. Do the shadows get baked into albedo and can the network then neural renderer recover from this?

On a similar note, it would be good to see some results of the physics based rendered images with target env maps that are used to supervise the neural renderer (Eq. 5). Also, the I_rendered|E_src would be interesting to see.

Most of the results that are shown are based on the source images captured on cloudy days or without strong directional light. While I realize that the results on very sunny days or even with sun glare will be worse, I would like to see how gracefully the proposed method degrades. Including some failure cases is always a plus in my mind.


**Questions**
- L137: relating to the comment in the *weaknesses*, how is the diffuse color supervised? Simply using the full RGB of the source images?
- In the environment modeling, why is the sky intensity a vector quantity? Is it a full image representation?
- In the physical rendering the buffers are mentioned to be 12 dim, but there is only position (3), normal (3), depth (1), ambient occlusion (1?). What do other dimensions represent?

**Limitations:**

The authors have extensively described the limitation and societal impacts of the proposed work in the supplementary material. I also appreciated the information about the GPU hours and proper acknowledgment of the data sources.

---

> ### Author Rebuttal · Authors · 2023-08-09
>
> Thank you for your thoughtful review. We are excited that LightSim is recognized as a “well-designed system” which is “technically sound” with “really thorough experimental evaluation” and “the design choices are well supported”. We are also glad that the reviewer believes the motivation is “very clear” and likes the idea of “neural deferred renderer”. We now address comments.
>
> **Q1: Simplification of reconstruction (i.e., diffuse color, material)** \
> **A1:** As mentioned in our limitations (supp. L343-345) we agree with the reviewer that LightSim makes simplifications in the reconstruction process and that improvements in intrinsic decomposition can further enhance performance. Recent concurrent works make steps towards better decomposition (FEGR [1], UrbanIR [2]), but it is still a challenging open problem to recover perfect decomposition of materials and light sources for large urban scenes (see **Figure R1-Right** of intrinsic decompositions of recent works - the recovered materials lack little semblance to semantic parts in the original scene). Additionally, these recent relighting works [1-4] also have shadows baked into the recovered albedo (**Figure R1-Left**).  **Figure R2** includes examples of the recovered albedo for four scenes from LightSim, and we also have baked shadows (mentioned in limitations, supp. L339-342). As discussed in L221-226, our novelty is in leveraging neural deferred rendering to overcome the limitations of purely physics-based rendering when the decomposition is not perfect. This allows us to generate better relighting results than prior works that have imperfect decompositions. It is an exciting future direction to incorporate better material decomposition along with neural deferred rendering for improved relighting.
>
> **Q2: Temporal consistency** \
> **A2:** We refer the reviewer to the relighting / shadow editing videos (3min 08s to 3min 51s) in our supplementary video. While we do not guarantee temporal consistency, LightSim can produce temporally consistent lighting simulation videos in most cases, therefore we did not explore other techniques to enforce temporal consistency explicitly. We believe the temporal consistency in neural deferred rendering comes from temporally and multi-view consistent inputs during inference (real image, G-buffers) as well as our combination of simulation and real paired relighting data during training.
> We believe further improving the temporal consistency is an interesting direction for future work.
>
> **Q3: Recovering the env map** \
> **A3:** Thanks for the suggestion. We believe optimization of the environment map would be an exciting direction to improve relighting. We explain our design choice in the following two aspects. (a) We use a feed-forward network for lighting estimation which is more efficient and can benefit from learning from a larger set of data. In contrast, the optimization paradigm is more expensive which requires per scene optimization but might be more accurate as mentioned by the reviewer. (b) The ill-posed nature of lighting estimation and extreme intensity range make inverse rendering challenging for outdoor scenes. Optimization of the environment map requires a differentiable renderer and high-quality geometry/material to achieve good results. The existing / concurrent state-of-the-art works [1, 3] still cannot solve the problem accurately (e.g., cloud occlusion example mentioned by the reviewer) as shown in Rebuttal Figure R1 (right bottom).
>
> **Q4: Novelty** \
> **A4:** See **General Response**.
>
> **Q5: Examples of view-independent reconstruction, I_rendered | E_src, strong directional light - failure cases** \
> **A5:** Thanks for the suggestions. We included those examples in Rebuttal Figure R2 (view-independent reconstruction), R3 (blender renderings) and R4 (figure cases). We will include those results and more analysis in the next version.
>
> **Figure R2:** \
> We provide examples of view-independent reconstruction for LightSim. Since we adopt the original RGB as the supervision to train the neural fields that map 3D location to the diffuse $\mathbf{k}_d$, the diffuse reconstruction results have the shadow baked.
>
> **Figure R3:** \
> The original RGB, diffuse reconstruction,  $\mathbf{I}_\mathrm{render} | \mathbf{E}^{\mathrm{src, tgt}}$, relit image and lighting reference are provided. We generate paired synthetic data under different lighting conditions and design a mixed sim-real training scheme.
>
> **Figure R4:** \
> We highlight two examples of LightSim applied to scenes with strong directional lighting and high sun intensity.  Each row shows the shadow editing / relighting results under 4 different sun angles of the target environment map. In Row 1, LightSim cannot fully remove source shadows in bright and sunny conditions (as mentioned in limitations) due to the baked shadows in the view-independent reconstruction, but with neural deferred rendering, LightSim still generates reasonable results. In Row 2, we depict a source image with high sun intensity and glare relit to a new target lighting. It is challenging to remove the sun glare and alter the over-exposed regions in this setting, but we can still apply some relighting effects to the cars and buildings in the scene (see arrows).
>
> **Q6: Why is the sky intensity a vector quantity?** \
> **A6:** Sky intensity contains 3 channels for R, G, B.
>
> **Q7: G-buffer dimension: position (3), normal (3), depth (1), ambient occlusion (1)** \
> **A7:** Yes, the buffers are 8 dimensions instead of 12 (in implementation, we loaded depth and ambient occlusion as 3-dim. images for simplicity). We will revise it in the next version.
>
> **Q8: "*relightable digital twins*" claim** \
> **A8:**  We will revise the term to be “*lighting-aware digital twins*” instead to indicate that the reconstructions also include scene lighting estimations.
>
> [1] FEGR. Wang et al., 2023. \
> [2] UrbanIR. Lin et al., 2023. \
> [3] NeRF-OSR. Rudnev et al.., 2022. \
> [4] Self-OSR. Yu et al., 2020.

---

> > ### Author Response · Authors · 2023-08-17
> > **Looking forward to follow-up discussions!**
> >
> > We thank the reviewer for taking precious time in checking our responses. We hope our answers and additional results address your concerns well. Specifically,
> >
> > - Q1/A1 and Q3/A3: Thanks for the suggestion. We explained the reasons for the simplifications of digital twin reconstruction. Better inverse rendering (base color, material, and lighting decomposition) is a promising direction that merits future study (see **Rebuttal Figure R1**).
> > - Q2/A2: While we do not guarantee temporal consistency, Lightsim can generate temporally consistent lighting simulation videos in most cases.
> > - Q4/A4: We clarified the novelty of LightSim.
> > - Q5/A5: We provided additional results in **Rebuttal Figure R2 (view-independent reconstruction), R3 (blender renderings) and R4 (figure cases)**.
> > - Q6/A6 and Q7/A7: We clarified the details of the sky intensity vector and G-buffer dimension.
> > - Q8/A8: We will change the term "*relightable digital twins*" to "*lighting-aware digital twins*".
> >
> > Please let us know if you have any additional or follow-up questions. We will be more than happy to clarify them. Any follow-up discussions are highly appreciated!

---

> > > ### Comment · Reviewer_5Bwf · 2023-08-17
> > >
> > > Thank you for the detailed responses, and I am sorry for my late reply.
> > >
> > > ***Temporal consistency***
> > >
> > > The argumentation that the temporal consistency emerges due to the temporarily consistent inputs is a good point, and it would be worthwhile mentioning it in the paper.
> > >
> > > ***Recovering the env map***
> > >
> > > While I agree that the optimization aspect also has its downsides, it at least has the potential to recover the sun location even if not observed. I am not sure if that is the case for the proposed inpainting network, do you maybe have any example of this? Maybe a hybrid approach of inference + short test-time optimization is the way to go.
> > >
> > > ***Failure cases with strong directional light***
> > >
> > > I think that it would be good to include these cases in the revised version of the paper and discuss this more concretely. It is a challenging problem to remove the strong shadows and other works also struggle with that so it doesn’t surprise me, but it is good to educate readers about the problem and to be honest about the limitations.
> > >
> > >
> > > My claim was not that using existing algorithms means that there is no novelty, but I still think that the novelty is on the lower side, as there are no significant new ideas that make or break the system. This being said, I really appreciate the added discussion and new visualizations/results. If the authors agree to include these in the revised version, I am willing to increase my rating to borderline accept.

---

> > > > ### Author Response · Authors · 2023-08-18
> > > > **Response to Reviewer 5Bwf**
> > > >
> > > > Thank you for the thoughtful suggestions and follow-up discussions.
> > > >
> > > > **Temporal consistency**: We will provide discussions about the temporal consistency in the revised manuscript.
> > > >
> > > > **Recovering the env map**: We agree that per-scene optimization has the potential to  recover more accurate scene lighting from partial observations. Like the reviewer, we believe future work combining prediction (based on prior knowledge) with test-time optimization (based on observation) may achieve superior performance, similar to works in other fields  [1, 2]. Note that in LightSim, we use the GPS and time data to get the approximate sun location. This enables us to recover the sun's location even if not observed (depending on the scene, we may not observe the sun directly from limited-FoV camera images), and the surrounding observed sky and scene context can still be used to approximately estimate the sun intensity. We will include more examples in the next version (We will send Rebuttal Fig. R7 to ACs via anonymous link if allowed).
> > > >
> > > > [1]  Learning to Reconstruct 3D Human Pose and Shape via Model-fitting in the Loop. Kolotouros et al., ICCV 2019. \
> > > > [2] AutoRF: Learning 3D Object Radiance Fields from Single View Observations. Müller et al., CVPR 2022.
> > > >
> > > > **Failure cases with strong directional light**: Thanks for the suggestion. As mentioned in the rebuttal, we will include the rebuttal figures (Figure R2 to R4) and provide more discussions about the limitations.
> > > >
> > > > **Novelty**: Thanks for the comments. The goal of this paper is to convey the importance of leveraging all available real data (via digital twins) and encourage further research in  exploiting digital twins for lighting simulation. Towards this goal. we propose a “novel data pair training scheme” [Reviewer 6GXP] that leverages the digital twins built from real world to generate synthetic paired images under different lighting conditions. Those synthetic data are then combined with real data to train the neural deferred rendering network. As demonstrated in the rebuttal figures, this is one of the key ideas that makes the system work. Additionally, different from existing works that conduct lighting estimation from single images, we fuse all available data (i.e., all six cameras and sun angles based on time and GPS) to reduce ambiguity.
> > > > Importantly, our paradigm is orthogonal to concurrent inverse rendering works (FEGR, UrbanIR), and one can expect further enhancements by marrying those works together.
> > > >
> > > > Thanks again for the prompt and thoughtful reply. Please let us know if there are any additional comments or questions.

---

> > > > > ### Author Response · Authors · 2023-08-21
> > > > > **Thanks for your review and post-rebuttal feedback**
> > > > >
> > > > > We thank the reviewers for agreeing to raise the score to borderline accept! Since the discussion period is coming to an end, please let us know if you need anything else from us.

---

### Author Rebuttal · Authors · 2023-08-09

We thank the reviewers for their thoughtful reviews and valuable comments. We are excited that the reviewers found our approach “novel” [**Reviewer 5Bwf, Reviewer 25CV, Reviewer ENZ6, Reviewer 6GXP**], and acknowledged our evaluation is “thorough” [**Reviewer 5Bwf**], the results are “good/high-quality” or have “noticeable improvements” over prior arts [**Reviewer 25CV, Reviewer ENZ6, Reviewer 6GXP, Reviewer G49E**].

In the following, we briefly summarize a few points. Please see individual responses for more details.

**Novelty [Reviewer G49E, Reviewer ENz6]**

Our paper’s novelty lies in developing a neural lighting simulation system for self driving, which is critical for thorough evaluation and more robust training of robot autonomy  before safe deployment. To our knowledge, we are one of the first to perform relighting for dynamic urban scenes (SDV and other actors moving in the scenario). LightSim significantly outperforms the state of the art, producing high-quality photorealistic driving videos under a wide range of lighting conditions.

We strongly believe LightSim is a critical and innovative step towards realistic and scalable lighting simulation for robotics. Through this paper, we also hope to convey the importance of leveraging available real data (digital twins) and propose a new regime of exploiting digital twins for lighting simulation.

While some individual components (reconstruction for dynamic scenes [75], environment map in-painting [59] and LDR-HDR lifting [69], G-buffers to aid learning [52]) are studied before, why they are used and how they are used are carefully designed for self-driving simulation. Specifically, different from existing works that conduct lighting estimation from single image (limited FoV) which is ill-posed and challenging, we fuse all available data (i.e., all six cameras and sun angles based on time and GPS) to reduce ambiguity (see Table A4). To overcome the lack of real world driving scenes captured under different lightings, we propose a “novel data pair training scheme” [**Reviewer 6GXP**] that leverages the digital twins that are built from real world to generate synthetic paired images under different lighting conditions. Those generated diverse synthetic lighting variation data are then combined with real data to train the neural deferred rendering network. The resulting framework is generic, interpretable and has various capacities including actor insertion, removal, modification, and rendering from new viewpoints, all in lighting-aware manner. We believe LightSim is not just a simple extension or integration of previous works.
Also, exploiting existing algorithms to realize a novel idea does not mean there is no technical contribution [**Reviewer 5Bwf and 6GXP**]. We hope the reviewers, in particular **Reviewer G49E**, can acknowledge this.

**Comparison to FEGR [Reviewer ENZ6, Reviewer 25CV]**

FEGR is a *concurrent and independent* work (Sec 2 Line 98 to Line 106). It was first made publicly available on arXiv on April 6., **less than two month of NeurIPS submission deadline May 11**. According to NeurIPS policy, “Authors are not expected to compare to work that appeared only a month or two before the deadline.” Therefore, the existence of FEGR should not weaken the novelty of LightSim. Moreover, as clearly stated in related work, while FEGR and LightSim are dealing with a similar task, the methods are very different. FEGR aims to conduct inverse rendering (intrinsic decomposition) from a single scene with strong regularization and priors. In contrast, LightSim proposes a “novel data pair training scheme” [Reviewer 6GXP]  and learns on many driven scenes to bypass the challenges of inverse rendering. We believe these two approaches are complementary and the combination of both can lead to a better system. LightSim also produces more realistic relighting videos for **dynamic scenes** and demonstrates the effectiveness in downstream detection.

**Please see the attached pdf for all rebuttal figures (Figure R1-6)**. We hope our responses and additional results can address the concerns. We look forward to follow-up discussions.

---

### Decision · Program_Chairs · 2023-09-21

**Decision:**

Accept (poster)

**Comment:**

This paper proposes a neural lighting simulation system for self-driving. Although the initial reviews are mixed, the authors provide a strong rebuttal and address most of the concerns raised by reviewers. The final assessments are generally positive, with four reviewers recommending acceptance and one reviewer with "borderline reject" indicating his/her willingness to raise the score to "borderline accept" in the comment. Aligned with the consensus of the reviewers, the AC thinks this paper makes good technical progress and thus recommends acceptance.